# Enhanced Land Subsidence Interpolation through a Hybrid Deep Convolutional Neural Network and InSAR Time Series

Zahra Azarm[1]; Hamid Mehrabi[1]; Saeed Nadi[2]

[1] Department of Geomatics Engineering, Faculty of Civil Engineering and Transportation, University of Isfahan, Isfahan, Iran

[2] Department of Civil an Environmental Engineering, Carleton University, Ottawa, Canada

*Correspondence to*: Hamid Mehrabi (h.mehrabi@eng.ui.ac.ir)

**Abstract**- Land subsidence, gradual or sudden, poses a significant global threat to infrastructure and the environment. This study introduces a hybrid approach that combines deep convolutional neural networks (CNN) with persistent scatterer interferometric synthetic aperture radar (PSInSAR) to estimate land subsidence in areas where PSInSAR data are unreliable or sparse. The proposed method trains a deep CNN using subsidence driving forces and PSInSAR data to learn spatial patterns and predict subsidence values. Our evaluation demonstrates that the CNN effectively mitigates discontinuities in PSInSAR results, producing a continuous and reliable subsidence surface. The model's performance was assessed using training, validation, and testing datasets, achieving root mean square errors (RMSE) of 3.99 mm, 8.47 mm, and 9 mm, respectively. In contrast, traditional interpolation methods such as Kriging, inverse distance weighting (IDW), and radial basis function (RBF) interpolation yielded RMSE values of 61.60 mm, 66.21 mm, and 61.76 mm, respectively, on the test dataset. Additionally, the coefficient of determination ($R^2$) for CNN, Kriging, IDW, and RBF was 0.98, -0.06, -0.22, and -0.06, respectively. The deep CNN model demonstrated an 85% improvement in subsidence prediction accuracy compared to conventional interpolation methods, highlighting its potential for accurate and continuous land subsidence estimation.

*Index Terms*— **Convolutional Neural Network (CNN); Prediction of subsidence; PSInSAR; driving forces; Kriging interpolation.**

## 1. Introduction

The gradual decrease in the height of the earth's surface, which is accompanied by slight horizontal displacements, is called subsidence. Due to the gradual nature of land subsidence, this phenomenon is also called "silent earthquake". Its harmful effects appear over a long period of time and carry significant risks. However, land subsidence is a global threat to urban areas around the world (Sun et al., 2023). This issue is an important global concern and is not limited to one region. Iran is facing an increasing challenge especially in this field. Human activities, such as mining and excessive underground water extraction, contribute to this problem. To address it, effective groundwater management to prevent unauthorized water extraction would help. However, land subsidence is not only caused by human actions, natural factors also play an important role. These include water table fluctuations, soil characteristics, depth of the bed rock, terrain features like elevation and aspect, vegetation cover, and prevailing climate. All these factors together create a complex landscape of land subsidence occurrences.

Precise leveling and GNSS observations offer high precision in measuring subsidence. Still, they are limited in their ability to investigate subsidence over a wide area due to their reliance on measuring sparse stations. These methods require multiple measurements at different locations, making it difficult to monitor subsidence over large areas (Fialko et al., 2005; Hu et al., 2012). On the other hand, Interferometric Synthetic Aperture Radar (InSAR), has emerged as a high spatial resolution and cost-effective technique for monitoring subsidence on a large scale (Chang et al., 2010; Tamburini et al., 2010; Tomás et al., 2011; Rucci et al., 2012; Amighpey & Arabi, 2016; Biswas et al., 2018; Gonnuru & Kumar, 2018; Khorrami et al., 2019). InSAR uses radar waves to carefully monitor changes in the Earth's crust surface over time. Methods that analyse radar images over time, known as time series analysis, make them very effective for monitoring subsidence, which usually occurs gradually over time. Persistent Scatterer Interferometric Synthetic Aperture Radar (PSInSAR) is particularly valuable for monitoring urban land subsidence. This is because there are many high-density Persistent Scatterer (PS) points, mainly associated with buildings and man-made structures. This abundance significantly improves the quality of the data within interferograms (Gao et al., 2019). Although these advantages are significant, dealing with the sparse and uneven distribution of PSs in both spatial and temporal dimensions is a significant computational challenge. The PSInSAR approach generates discontinuous results,

as it calculates subsidence exclusively at PS points. Consequently, it becomes imperative to employ intelligent
interpolation instead of mathematical or stochastic methods, between these data points to fill out these gaps (Naghibi
et al., 2022).
Subsidence is a complex physical phenomenon influenced by a multitude of factors, such as changes in groundwater
levels, soil type, bedrock depth, slope, elevation, Aspect, etc. To obtain the subsidence in the whole area, interpolation
methods between PSs and artificial intelligence methods (which are trained with features affecting subsidence) can be
used. Interpolation methods between PSs and artificial intelligence methods (trained with features affecting
subsidence) can be used to obtain subsidence in the entire area. Classical interpolation methods (e.g. Kriging, IDW,
RBF (Mehrabi & Voosoghi, 2018), RMLS (Mehrabi & Voosoghi, 2015)) do not consider the physics of the issue,
making their results less reliable. So, it is very important to apply methods that take into account the real characteristics
of the phenomenon, especially when monitoring the subsidence. Recently, machine learning methods specifically deep
convolutional neural network (CNN) shows encouraging results in various applications. In the larger context of land
subsidence prediction models, we find two main categories: Physical Process Models: These models simulate
subsidence by incorporating factors like geotechnical mechanics, soil properties, and water dynamics. They are
frequently used in large-scale projects but require a substantial amount of prior knowledge and data (Nie et al., 2015);
Mathematical or Statistical Models: These models predict subsidence based on historical elevation data and past trends
(Zhu et al., 2010).
Several studies have investigated various forecasting models, methodologies, and influencing factors to improve our
understanding of this field. Neural networks have emerged as powerful prediction tools, so neural networks have been
used in the field of subsidence prediction using its driving forces. (Zhu et al., 2010; Azarakhsh et al., 2022; Ku & Liu,
2023). Lee et al. (2023) employed data from an urban area in Korea to develop a machine learning-based model for
predicting land subsidence risk. Their methodology incorporated historical land subsidence data along with attribute
information pertaining to underground utility lines in the specified region. The research team utilized machine learning
algorithms such as Random Forest (RF), eXtreme Gradient Boosting *(*XGBoost), and Light Gradient Boosting
Machine (LightGBM) for the analysis and prediction of land subsidence risks (Lee et al., 2023). Sadeghi et al. (2023)
combined full consistency decision-making (FUCOM) and GIS methodologies to assess Iran's vulnerability to land
subsidence. Their approach resulted in the development of a hierarchical FUCOM-GIS framework, which highlighted
critical factors such as water stress, groundwater depletion, soil type, geological time scale, and rainfall amount as the
main drivers of land subsidence. Researchers commonly validate their results by comparing them with InSAR
analyses, identifying areas exhibiting notable subsidence. Furthermore, the research assessed the risks to power
transmission lines and substations, revealing structural issues such as pier sinking, electric insulator deviation, and
cracking(Sadeghi et al., 2023). In another study focused on Dechen County, China, Wang et al. (2023) employed
Backpropagation Neural Network (BPNN) and RF algorithms, in conjunction with various monitoring data sources,
GIS, and SBAS technology, to predict trends in land subsidence. Their findings underscored Sugianto town as the
most severely affected area, with an annual average subsidence rate of -40.71 mm per year. The study highlighted that
changes in both deep and shallow groundwater levels were the primary drivers of land subsidence in this region.
Notably, the BPNN model demonstrated higher prediction accuracy compared to the RF model, especially when
considering changes in groundwater levels (Wang et al., 2023). Furthermore, Zhuo et al. (2020) demonstrated that the
integration of the GM (1,3) model with neural networks and ground-related variables shows great potential for
achieving highly accurate subsidence predictions. The proposed approach has the
capability to replace traditional precise leveling methods in long-term subsidence forecasting, offering valuable
insights for urban disaster prevention (Zhou et al., 2020).
Deng et al. (2017) conducted research on the integration of PSInSAR with Grey system theory for monitoring and
predicting land subsidence, as demonstrated in the Beijing plain (Deng et al., 2017). Precision mapping of complete
subsidence basins faces challenges, especially when dealing with image pairs with limited temporal separation. Rapid
deformations and vegetative changes in such scenarios introduce complexity. Strategies, such as combining
differential interferometric synthetic aperture radar (DInSAR) with the probability integral model (PIM), have been
introduced to effectively delineate subsidence basins resulting from mining activities (Fan et al., 2015).
The remarkable effectiveness of the RF model in mapping the susceptibility of land subsidence deserves attention.
This approach demonstrates exceptional capabilities in identifying key factors that contribute to subsidence
occurrences, such as the proximity to faults, elevation, slope angle, land use patterns, and water table levels. These
factors play a crucial role in influencing the likelihood of subsidence events (Mohammady et al., 2019). In addition,
the integration of fuzzy logic techniques and neural networks has been used to predict subsidence (Ghorbanzadeh et
al., 2020).
Land subsidence is a significant geological risk and predicting and investigating this phenomenon is vital. Traditional
monitoring and forecasting methods have limitations and require more advanced approaches. Kumar et al. (2022)
utilize recurrent neural networks (RNNs), specifically Vanilla and Stacked Long Short-Term Memory (LSTM)
models, to forecast land subsidence in the Jharia Coalfield, Dhanbad, India. Using data from 14 locations collected
through the Modified PSInSAR technique, the study shows these models can effectively predict deformation rates,
identifying critical subsidence levels at Nai-dunia basti, Digwadih, and Godhar. This research underscores the
potential of integrating advanced monitoring techniques with sophisticated predictive models to better anticipate and
mitigate land subsidence impacts (Kumar et al., 2022).
The integration of InSAR processing with deep learning methods in modeling and predicting land subsidence has
shown significant promise. This approach demonstrates substantial capabilities in identifying and predicting
subsidence in regions around Lake Urmia by leveraging Sentinel-1 data and small baseline subsets (SBAS) InSAR
methods. Key factors such as rainfall, groundwater levels, and lake area variations, measured using TRMM, GRACE,
and MODIS satellite data, were critical in understanding subsidence dynamics. Moreover, the application of machine
learning models, including multi-layer perceptron (MLP), convolutional neural network (CNN), and long short-term
memory (LSTM) networks, has been instrumental in improving prediction accuracy. The ensemble model combining
these networks outperformed individual models, achieving enhanced prediction reliability (Radman et al., 2021).
Predicting deformation is essential for early detection of abnormal conditions and timely intervention. A recent study
introduced a deep convolutional neural network (DCNN) approach to forecast time-series deformation using InSAR
data. The research, conducted at Hong Kong International Airport, demonstrated that the DCNN could effectively
predict both linear land settlement and nonlinear thermal expansion of structures with high accuracy. The study's
findings highlight the DCNN's potential to enhance early warning systems by providing precise short-term
deformation predictions, thus enabling better risk management and mitigation strategies (Ma et al., 2020).
In this study we used a CNN model trained over the area where subsidence is available through PSInSAR. Then this
model is used over other areas where subsidence cannot be obtained from PSInSAR processing. The proposed method
follows three main steps: Calculation of subsidence in PSs by PSInSAR method, calculation of subsidence driving
forces, and training CNN.
**2. Methodology**
**2.1 PSInSAR**
PSInSAR is a remote sensing technique that utilizes SAR images to monitor surface deformation over time. It relies
on identifying PSs, which are stable points on the Earth's surface reflecting radar signals consistently. PSInSAR
combines multiple interferograms created by comparing SAR images of the same area taken at different times. By
analysing phase differences between radar signals in these interferograms, it detects changes in the Earth's surface
over time. PSInSAR has significant advantages over DInSAR, as it effectively eliminates topographic errors,
atmospheric noise, and addresses temporal and spatial correlation issues between radar images (Ferretti et al., 2001;
Wasowski & Bovenga, 2014; Gonnuru & Kumar, 2018). PSInSAR, a form of differential interferometry, involves
analysing a collection of at least 15 SAR images captured at  different times, all covering the same area (Crosetto et
al., 2016). PSInSAR finds diverse applications, including monitoring subsidence in urban areas (Ferretti et al., 2000;
Luo et al., 2013) and tracking natural hazards such as landslides, earthquakes, and volcanic
activity (Peltier et al., 2010). However, one drawback of PSInSAR is the lack of continuity between PSs, as they
depend on the land use of the area. These PSs are more abundant in areas  with buildings, dams, oil wells, pipelines,
electric fences, roads, rocks, and bridges (Din et al., 2015), but they are relatively scarce in vegetated areas.
Consequently, PSInSAR performs best in urban areas and regions with rocky terrain (Oštir & Komac, 2007).
In this article, the amplitude dispersion index is used to select the persistent scatterer points, Eq. (1). The usual
threshold of the amplitude dispersion index is limited between 0.2 and 0.4 (Conway, 2016).
$D_A = \sigma_A / \mu_A$ (1)
where $\mu_A$ , $\sigma_A$ are standard deviation and mean values of the radiometrically corrected amplitude of pixels. In PSInSAR
the amplitude data from SAR images is carefully examined to identify specific PSs while excluding those affected by
space-time decoherence and atmospheric delay (Li et al., 2004).

## 2.2. Deep Convolutional neural network (CNN)

CNNs, or Convolutional Neural Networks, are deep learning algorithms widely employed for various image-related
tasks such as image recognition, classification, and regression. They learn and extract essential features from raw
images by processing them through multiple layers of filters, known as "convolutions." This multi-layer processing
progressively extracts more abstract features. These filters are trained using backpropagation, a technique that adjusts
filter weights based on the difference between predicted and actual outputs. In addition to convolutional layers, CNN
typically includes pooling layers to down sample the convolutional output and fully connected layers to use the
extracted features for image classification. CNN has gained popularity, particularly after the success of AlexNet in
the ImageNet challenge in 2012 and has since become the dominant approach for image recognition tasks.
CNN is used in various fields, including medical imagery (Lee et al., 2017), classification (LeCun & Bengio, 1995),
segmentation (Nair & Hinton, 2010; Van Do et al., 2024), image reconstruction (Christ et al., 2016; Lakhani &
Sundaram, 2017; Elboushaki et al., 2020), and natural language processing (Kim et al., 2018). While CNN are often
associated with categorical tasks, they are also highly effective in regression tasks, where the goal is to predict
continuous output variables instead of discrete labels. In CNN regression, the network typically has a single output
neuron in the final layer that generates a continuous value instead of a probability distribution for classification. It is
important to note that CNN requires a lot of input data, especially for image processing. As the network's depth
increases, so does its complexity, resulting in a larger number of weight parameters, which can sometimes create
challenges during training (Liu et al., 2018). CNN introduced the concept of local connections between layers with
typical components including convolution, activation and pooling layers (Chen et al., 2018). The convolutional layer
learns image features from small sections of input data through mathematical operations involving the input image
matrix and a filter or kernel. The activation layer introduces nonlinearity into the network, commonly using the
Rectified Linear Unit (ReLU) function.
CNN regression is a valuable approach for predicting continuous output variables and has found applications in
various fields including geology and civil engineering. CNN regression can also be used to predict subsidence. By
training a CNN model with input-output pairs, where inputs are subsidence driving forces and outputs represent
subsidence values, researchers can predict subsidence at single-pixel levels and provide valuable insights.
To predict land subsidence, we trained a CNN regression model with the architecture shown in Figure 1. The CNN
has 31 layers, including three 1×1 convolutional layers, three 3×3 convolutional layers followed by three 2×2 max-
pooling layers, Batch Normalization layers, drop out layers with a rate of 0.1, and two fully-connected layers with
1024 Rectified Linear Unit (ReLU) activation neurons, two fully-connected layers with 512 ReLU activation neurons,
and a fully-connected layer with 256 ReLU activation neurons. The input dimensions are 30×30×9, where 30×30
patches separated from the neighborhood of each scattered point and 9 features are used as network input.

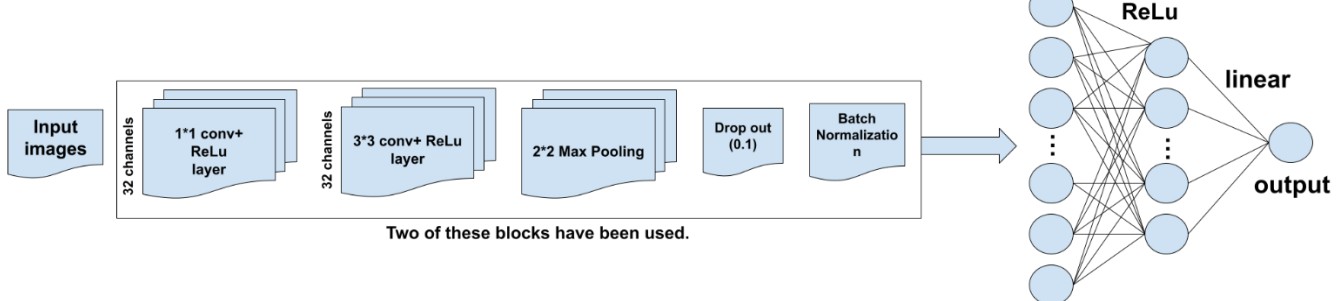


**Figure 1: Illustration of CNN Architecture**

**2.2.1. Hyperparameter Tuning Process**

After creating the model architecture, model inputs were normalized to a range of [0, 1] to ensure consistent input scaling, which is crucial for the stable performance of the neural network. Then we tuned the hyperparameters of the CNN regression model, including the loss function, optimizer, batch size, learning rate, activation function, and number of epochs. The best model was saved based on its performance metrics. The rationale for each hyperparameter is explained in detail and the optimal parameters for the model are given in Table 1:

- **Activation function:** We used the Rectified Linear Activation (ReLU) function in the hidden layers due to its effectiveness in mitigating the vanishing gradient problem and promoting sparse activations. For the output layer, a Linear activation function was employed to ensure the model could predict a continuous range of values.

- **Loss function:** We considered both Mean Squared Error (MSE) and Mean Absolute Error (MAE) as potential loss functions. MSE penalizes larger errors more heavily than MAE, making it suitable for scenarios where outliers significantly impact the model's performance. Given MSE's properties and its ability to improve the model's performance by reducing fluctuations and speeding up convergence, we selected MSE as our loss function. The MSE is calculated as follows:

$$MSE = {1}/{N} \sum_i^N (Y_i - \widehat{Y}_i)^2 \tag{2}$$

where $Y_i$ represents the actual values, $\widehat{Y}_i$ represents the predicted values, and $N$ is the number of observations.

- **Batch Size**: We experimented with batch sizes of 64 and 128. A larger batch size of 128 was chosen as it provided a good balance between training speed and model performance, allowing more stable gradient estimates.

- **Learning Rate**: The initial learning rate was set to 0.001, but we found that a smaller learning rate of 0.0001 led to more gradual and stable convergence, reducing the risk of overshooting the optimal solution.

- **Optimizer**: The Adam optimizer was selected for its adaptive learning rate capabilities and efficiency in handling sparse gradients. It combines the advantages of both the AdaGrad (Adaptive Gradient Algorithm) and RMSProp (Root Mean Square Propagation) algorithms, making it suitable for our regression task.

- **Number of Epochs**: We initially set 100 epochs but extended this to 150 epochs to ensure the model had sufficient time to learn the underlying patterns in the data without overfitting.

- **To divide the data**: we initially allocated 15% to the test data, 15% to the validation data, and 70% to the training data. However, we observed high-cost function fluctuations in the training and validation data. To mitigate this issue, we adjusted the data split to 80% for training and 10% each for testing and validation, which helped reduce the fluctuations

**Table 1. Key parameters of the CNN**

| parameters | value |
| --- | --- |
| Activation function of hidden layer, input layer | ReLu |
| Activation function of output layer | Linear |
| Input shape | 30×30×9 |
| Loss function | MSE |
| Batch size | 128 |
| Learning rate | 0.0001 |
| Epoch | 150 |
| Train-validation-test | 80% -10% -10% Of total data |

### 2.3. Driving forces of subsidence

The selected driving factors for predicting subsidence—NDVI, distance from wells, land use, water table map, altitude, slope, SPI, TWI, and aspect—are well-supported by extensive research and have been identified as significant

predictors in previous studies (Yang et al., 2014; Fan et al., 2015; Conway, 2016; Abdollahi et al., 2019; Andaryani et al., 2019; Mohammady et al., 2019; Zang et al., 2019; Ghorbanzadeh et al., 2020; Shi et al., 2020; Zhou et al., 2020; Zhao et al., 2021; Wang et al., 2023). By incorporating these factors into the subsidence prediction model, this study ensures a comprehensive approach that reflects the complexity of subsidence phenomena.

1. NDVI is a crucial indicator of vegetation health and land cover changes. Changes in NDVI can reflect alterations in land use practices, such as urbanization or agricultural expansion, which are closely linked to subsidence. Healthy vegetation typically reduces the need for excessive groundwater extraction, while barren or urbanized areas might correlate with higher subsidence due to increased groundwater use.

2. The distance from groundwater extraction wells is a critical factor in subsidence studies. Areas closer to high-density exploitation wells often experience more severe subsidence due to the localized impact of extensive groundwater withdrawal.

3. Land use changes, including urbanization, agricultural expansion, and deforestation, influence subsidence rates. Urban areas often experience higher subsidence due to increased groundwater extraction for residential, industrial, and agricultural purposes.

4. Groundwater level changes, as depicted in water table maps, are directly linked to subsidence. Over-extraction of groundwater leads to a drop in the water table, causing the ground to compact and subsidence. Groundwater depletion is a primary contributor to subsidence, emphasizing the importance of preventing unauthorized withdrawals and effectively managing water resources.

5. Altitude influences subsidence through its effect on hydrological processes. Altitude affects the distribution and movement of groundwater. Higher altitudes typically receive more precipitation, which can infiltrate the ground and recharge aquifers. At lower altitudes, reduced precipitation and higher evaporation rates can lead to a lowering of the water table. When groundwater is extracted faster than it is replenished, it can result in subsidence. The amount of water in the soil, influenced by altitude through precipitation and drainage patterns, affects soil compaction. High altitude areas with abundant rainfall can lead to saturated soils which are less prone to subsidence. Conversely, in lower altitude areas with less precipitation, soils may dry out and compact more easily, contributing to subsidence.

6.  Slope affects water runoff and infiltration rates. Steeper slopes may reduce infiltration, leading to less
groundwater recharge and potentially higher subsidence rates in adjacent flat areas.

7.  Aspect affects solar radiation and, consequently, evaporation and soil moisture levels. Different aspects can
lead to variations in vegetation cover and groundwater recharge, influencing subsidence. Additionally, the
slope and aspect of an area can influence drainage patterns, erosion, and sediment production, all contributing
to subsidence.

8.  The Stream Power Index (SPI) measures the power of water flow in depositing and causing soil erosion. As
a result, this index can be an important input for subsidence prediction models. The equation used to calculate
SPI is as follows (Pradhan et al., 2014):

$SPI = \alpha * \tan\beta$                                                                     (3)

Here, $\alpha$ flow accumulation, and $\beta$ represents the slope.

9.  The Topographic Wetness Index (TWI) is a mathematical formula that quantifies the effect of local
topography on the flow of surface water. It is a physically based index that can be used to determine flow
direction and accumulation and has many practical applications in fields such as hydrology, agriculture, and
geology. TWI indicates areas of potential soil moisture accumulation. Areas with high TWI values are likely
to have more groundwater recharge, which can mitigate subsidence.

In rainfall runoff modeling, TWI can be used to predict the amount and timing of runoff in a specific area,
while in soil moisture modeling it can be used to predict the spatial distribution of soil moisture. Overall, the
TWI is a useful tool for understanding and predicting the movement of water across the landscape (Qin et
al., 2011). Also, TWI identifies areas that can be affected by flooding from rainfall events (Ballerine, 2017).
TWI equation is as follows (Moore et al., 1991):

$TWI = \ln(\alpha/\tan\beta)$                                                                   (4)

Where $\alpha$ the upslope contributing area and $\beta$ is slope. TWI is calculated utilizing a Digital Elevation Model
(DEM) of the study areas.

It's essential to recognize that examining one factor alone is not enough to predict subsidence. A linear relationship
between groundwater level changes and subsidence may exist in certain regions, but this linear relationship does not
exist in all regions. Each region has unique characteristics such as soil type, fault lines and slope, etc. Subsidence is a
complex phenomenon that requires a comprehensive investigation that takes into account all relevant factors.
Therefore, thorough analysis is necessary to obtain a comprehensive understanding of subsidence in a particular area
(Azarm et al., 2023).
**3. Study Area and Datasets**
**3.1. Study area**
The studied area is in Isfahan province and includes the cities of Isfahan, Mahyar, Khomeinishahr and Falavarjan.
This region has a rich history of human habitation, diverse cultural heritage and a wide range of economic activities.
Covering approximately 7000 square kilometers, this area displays various uses, including urban, agricultural and
industrial areas. Its climate is semi-arid, characterized by hot summers and cold winters. The primary sources of water
in this area are the Zayandeh-Rud River and several underground aquifers that provide various uses such as agriculture,
drinking water, and industrial needs (Neysiani et al., 2022) (Figure 2). To effectively monitor and predict land
subsidence in this study area, we used advanced techniques such as radar interferometry and convolutional neural
networks (CNN). Our goal was to provide an accurate and reliable estimate of land subsidence in the study area by
integrating these techniques and considering complex subsidence driving forces.









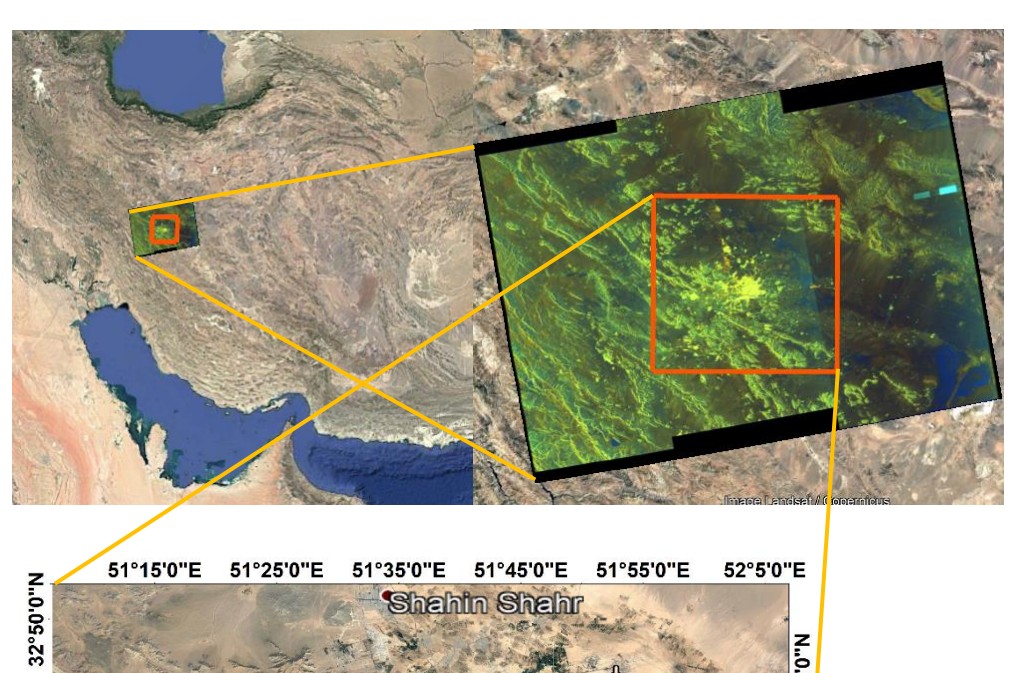

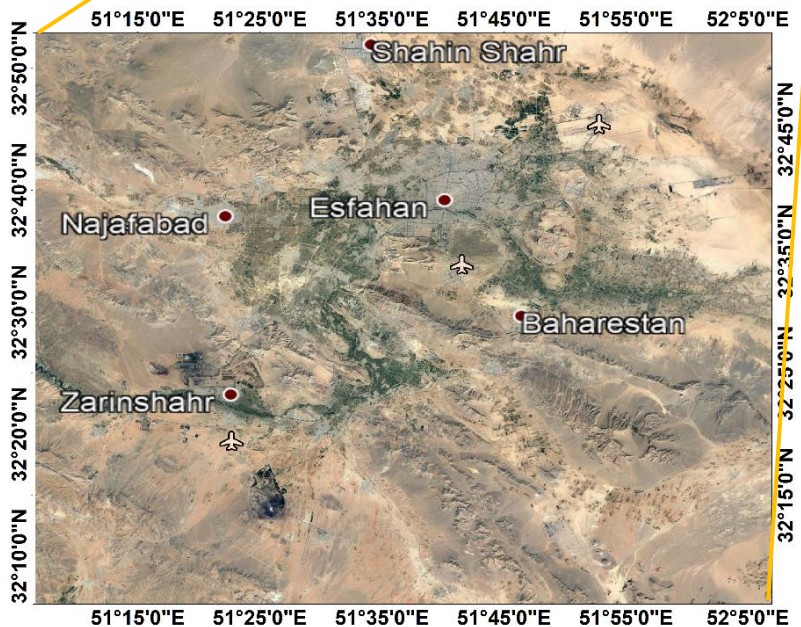

Figure 2: Geographic overview of the study area. (© Google Earth)

**3.2. Datasets**

**3.2.1. Sentinel-1A Radar Images**

This study utilizes 73 radar images from the Sentinel-1A satellite to analyze subsidence trends in the study area over

a seven-year period, from 2014 to 2020. The data was collected from the Ascending track 28. The Sentinel-1A satellite,

launched by the European Space Agency (ESA), operates in C-band and provides Synthetic Aperture Radar (SAR)

imagery with a spatial resolution of 5 meters by 20 meters. The images were acquired at six-day intervals, ensuring

high temporal resolution for detecting ground movements. The Interferometric Wide (IW) swath mode was used,

offering comprehensive coverage of the study area.

The PSInSAR technique was applied to the Sentinel-1A data using Sarproz software. This method is particularly
effective in urban and semi-urban areas where permanent scatterers are abundant. The precise processing steps
involved coregistration, interferogram generation, phase unwrapping, and geocoding to produce detailed subsidence
maps (Ferretti et al., 2001).
**3.2.2. Digital Elevation Model (DEM)**
The 30-meter Shuttle Radar Topography Mission Digital Elevation Model (SRTM DEM) was employed to calculate
various topographical and hydrological indices, including SPI, TWI, slope, and aspect. These indices were computed
using ArcMap software, providing essential insights into the terrain characteristics influencing subsidence (Fig.3,
Fig.4).

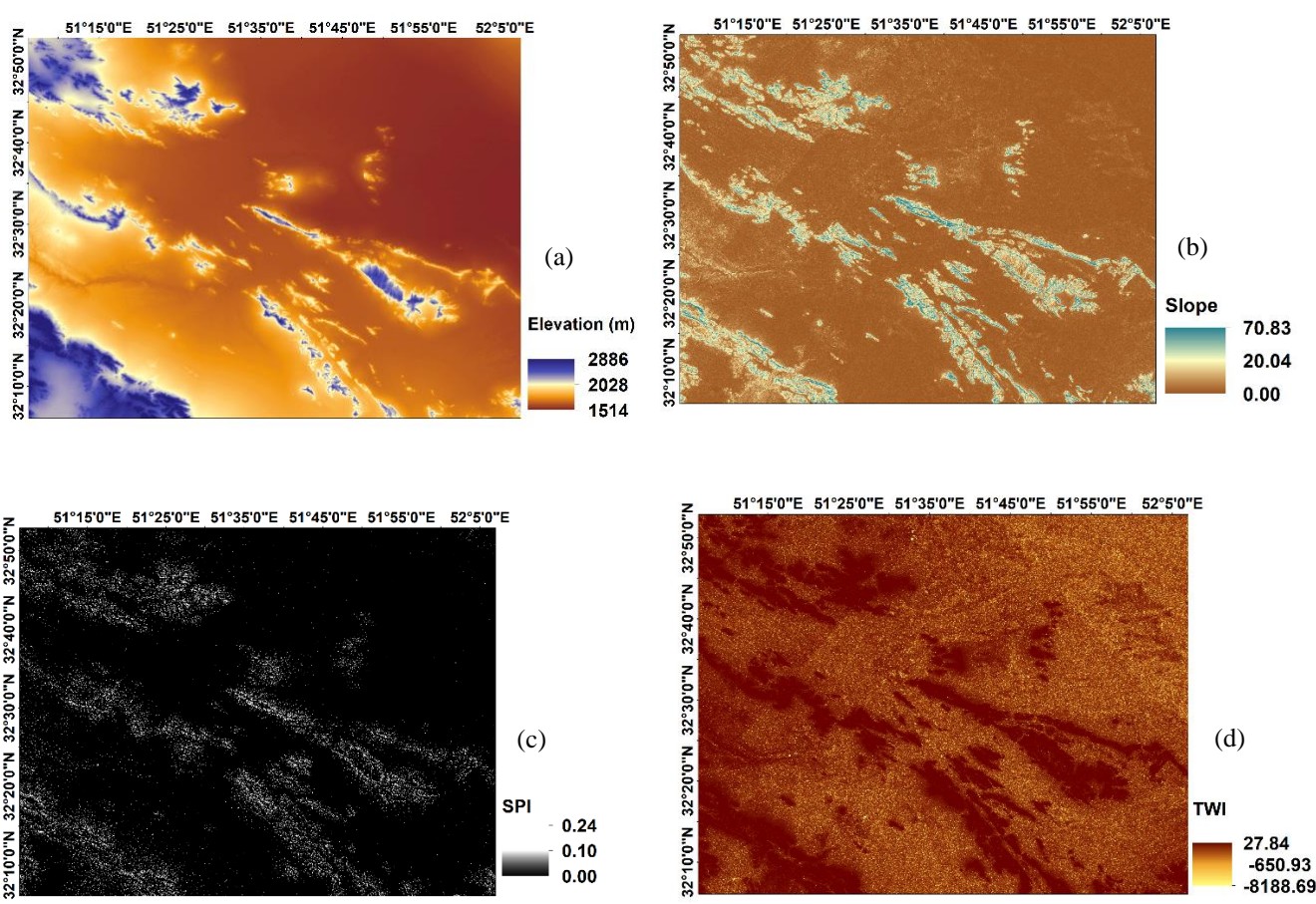

**Figure 3: Subsidence driving forces - (a) Elevation, (b) Slope, (c) SPI, (d) TWI**

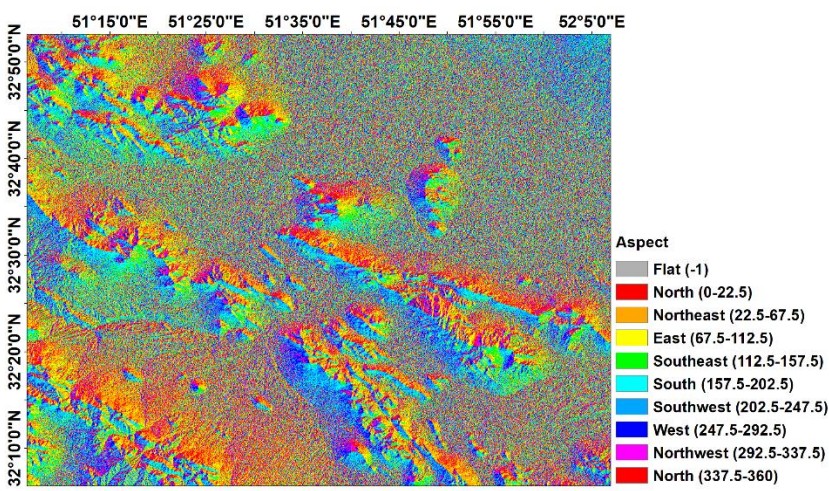


**Figure 4: Subsidence driving force – (e) Aspect**


### 3.2.3. Optical Satellite Images

Optical images from the Landsat 8 satellite, launched by NASA, were used to extract NDVI and land use information for the year 2020. The Landsat 8 images, with a spatial resolution of 30 meters, were processed using Envi software to calculate average annual changes in NDVI between 2014 and 2020. This analysis helps in understanding the impact of vegetation and land use changes on subsidence (Fig. 5).

### 3.2.4. Groundwater Monitoring Data

Groundwater level changes were investigated using data from piezometric wells within the study area. The groundwater monitoring data, covering the period from 2014 to 2020, were sourced from Isfahan Regional Water Authority. These data were collected monthly and provided detailed information on the groundwater table fluctuations. The data were processed to generate water table maps, which were then analysed in relation to subsidence patterns. In areas with high densities of exploitation wells, the probability of subsidence increases due to significant groundwater extraction. The distance from these wells was calculated and included as one of the driving forces for subsidence (Fig. 5).

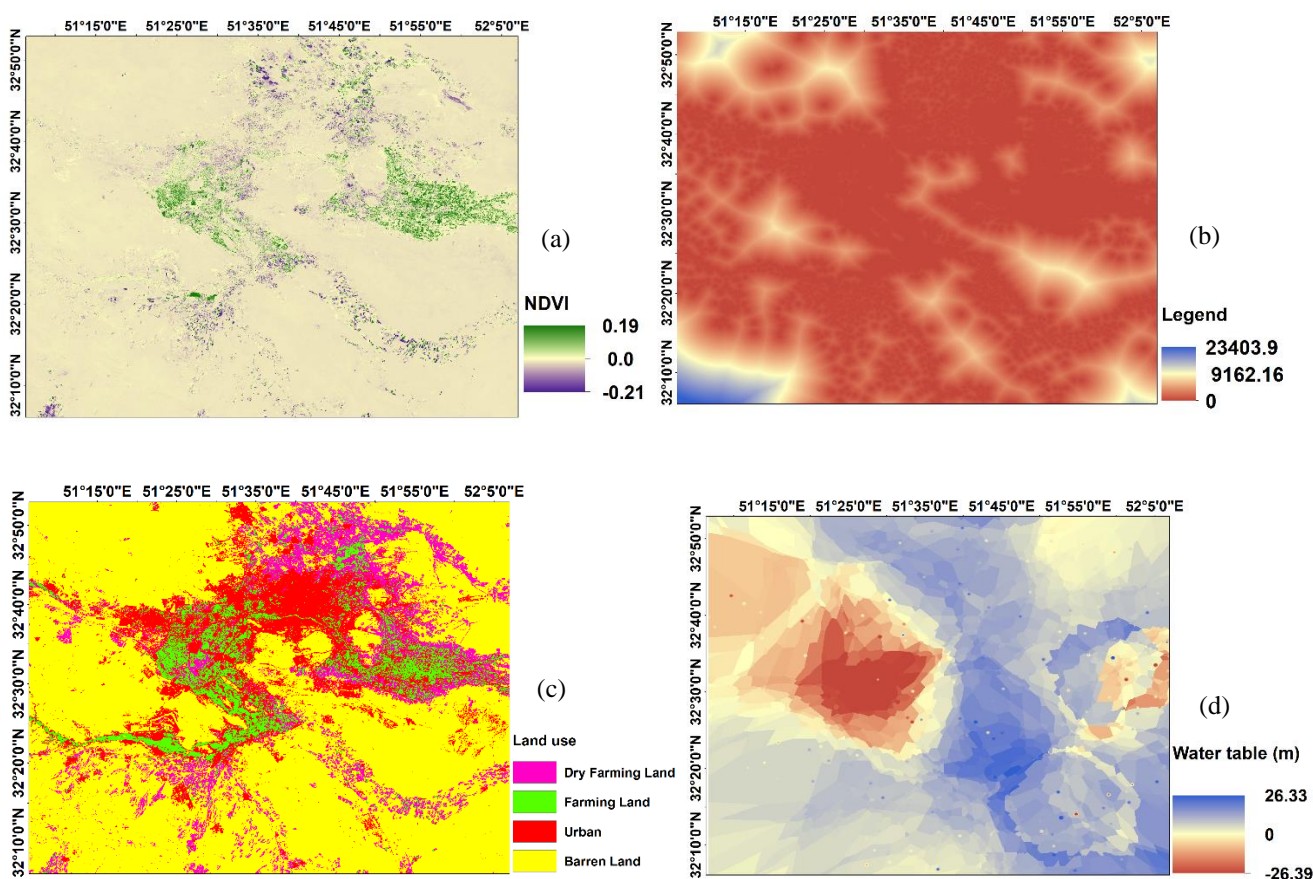

**Figure 5: Driving forces of subsidence- (a) NDVI (b) Distance of Wells (c) Land use, (d) Water table map in 2014 to 2020**

## 4. Result

### 4.1. Results of CNN

CNN was trained using the calculated driving forces and subsidence at the PSs and the performance of the network

by analysing the graphs of the cost function (mean squared error) for the training and validation data, as shown in

Figure 6, the RMSE values of this model for the training, validation and test data are 3.99, 8.47 and 9 mm, respectively.

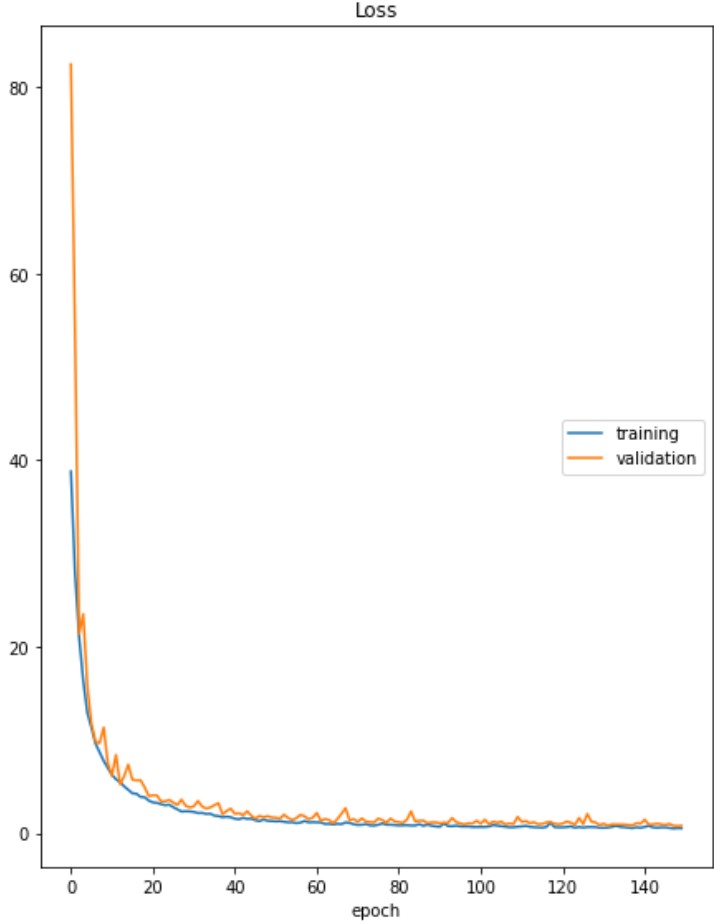

344

**Figure 6: Cost function of training and validation data**

346

**4.2. Comparison between CNN and traditional interpolation methods**

In our study, we employed four distinct methods to create a continuous subsidence surface: a Convolutional Neural

Network (CNN) and three traditional interpolation methods—Kriging, Inverse Distance Weighting (IDW), and Radial

Basis Function (RBF). The traditional interpolation methods were utilized to interpolate between Persistent Scatterers

(PSs) and calculate the subsidence across all pixels within the study area, based solely on the spatial distribution of

the PSs. However, these methods do not account for the subsidence driving forces, and their accuracy can be

compromised by irregular distributions of PSs.

In contrast, the CNN approach was trained using subsidence driving forces to predict subsidence and generate a

continuous subsidence surface. This method is particularly effective in handling irregularly distributed data points,

making it suitable for scenarios where PSs are unevenly distributed across the study area. By incorporating subsidence

driving forces, the CNN can provide a more reliable prediction of subsidence compared to the traditional interpolation

methods. To evaluate the accuracy of these methods in predicting subsidence, we used several performance metrics,
including the Root Mean Squared Error (RMSE), Mean Absolute Error (MAE) and R-squared (R2), The values of
these metrics for each method on the train and test data are given in Table 2. To further validate the superiority of the
CNN model, we conducted statistical significance tests. A t-Test was performed to compare the performance metrics,
with the results indicating a statistically significant improvement in the CNN model's performance over the traditional
interpolation methods (p-value < 0.05). These results indicate a statistically significant improvement in the accuracy
of the CNN compared to the traditional interpolation methods.
**Table 2 . Compare between interpolation methods to predict subsidence (Train data)**

| | Train data | | | Test data | | |
|---|---|---|---|---|---|---|
| **Method** | **RMSE (mm)** | **MAE (mm)** | **R2 score** | **RMSE (mm)** | **MAE (mm)** | **R2 score** |
| CNN | 3.99 | 2.18 | 0.99 | 9.06 | 3.69 | 0.98 |
| Kriging | 62.78 | 39.19 | -0.09 | 61.60 | 37.90 | -0.06 |
| IDW | 67.32 | 40.52 | -0.25 | 66.21 | 39.30 | -0.22 |
| RBF | 62.67 | 38.95 | -0.08 | 61.76 | 37.92 | -0.06 |


Error distribution maps are visual tools that illustrate the spatial distribution of prediction errors across the study area.
These maps play a crucial role in evaluating the performance of subsidence prediction models, such as the
Convolutional Neural Network (CNN) and traditional interpolation methods (Kriging, IDW, and RBF).
By plotting the differences between the predicted and PSInSAR subsidence values at various locations, error
distribution maps help identify patterns or areas where the models perform well or poorly. Clusters of high errors
indicate that traditional interpolation methods do not perform well in areas where the range of subsidence values is
greater than the average values of the entire study area and in areas with sparse PS distribution. These methods tend
to have the highest errors at these points, which are often critical for accurate subsidence assessment. In contrast, the
CNN demonstrates more consistent performance due to its training on subsidence driving forces, resulting in lower
errors in these high-variance regions.
In our study, the error distribution maps confirmed the findings from the quantitative performance metrics (RMSE,
MAE, and R² score). The CNN showed a more uniform error distribution, indicating its effectiveness in handling
irregular data distributions and incorporating subsidence driving forces. This visual evidence supports the conclusion
that the CNN provides a more reliable and accurate subsidence prediction compared to traditional interpolation
methods (Fig. 7).

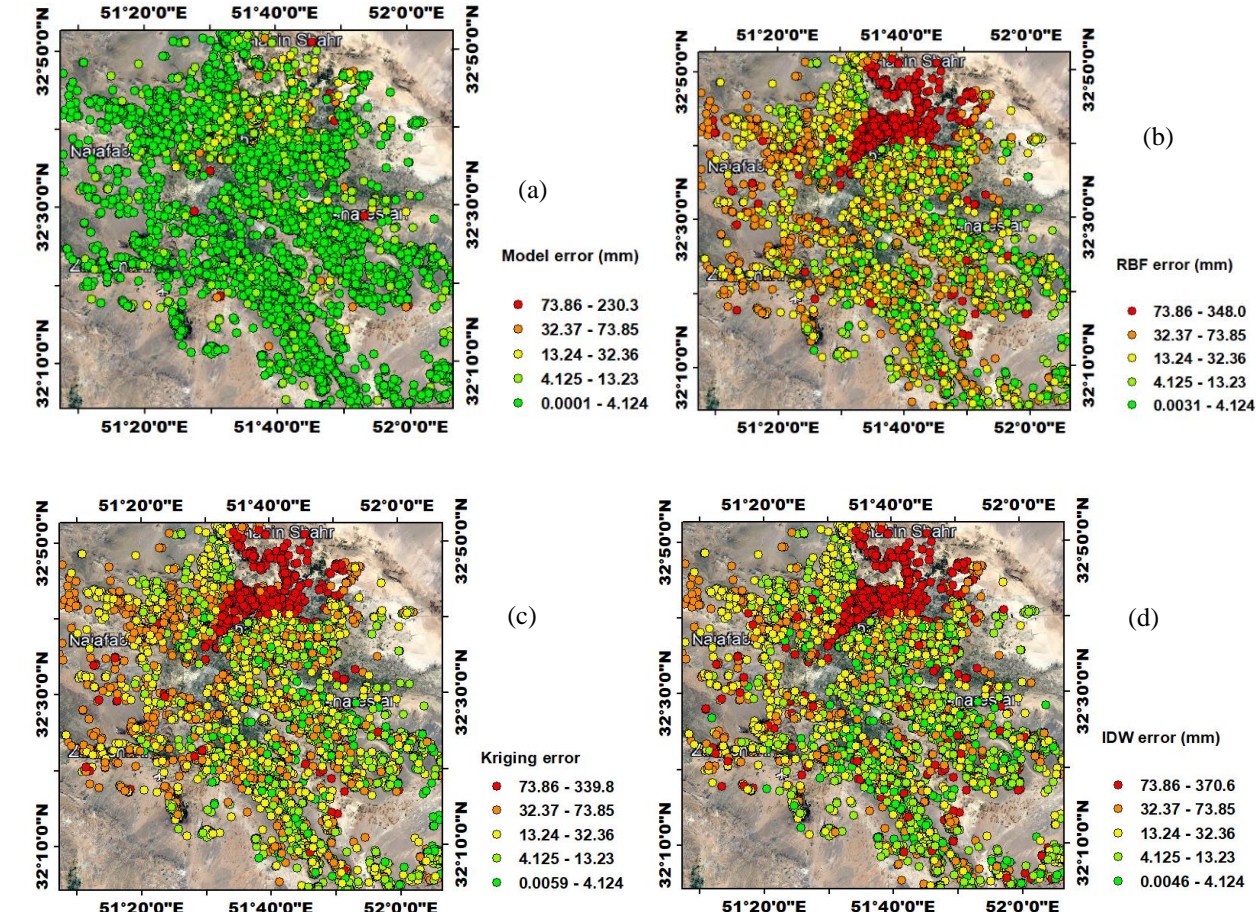

**Figure 7: Error distribution map of (a) CNN (b) RBF (c) Kriging (d) IDW**

## 4.3 Subsidence of Study area

In our analysis of land subsidence in the Isfahan region, we processed a total of 73 Sentinel-A images using the
PSInSAR method. Through this process, we identified PSs by applying a range amplitude dispersion index threshold
of 0.2 and a temporal correlation threshold of 0.8. The maximum velocity for these PSs was observed in the northeast
of the study area, specifically near Shahid Beheshti Airport in Isfahan, measuring at -67 mm per year. This significant
rate resulted in a cumulative displacement of approximately -33 cm in the period from 2014 to 2020 (Fig. 8).

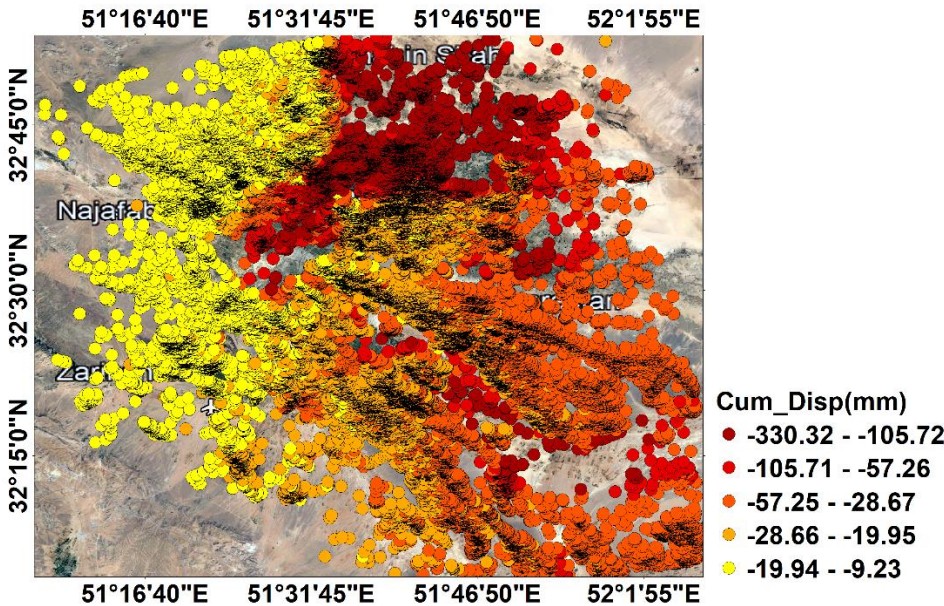


**Figure 8: Cumulative displacement of PSs in 2014 to 2020**

A velocity map was created using Kriging interpolation between PSs. The results showed that the highest velocity,

approximately 67 mm per year, was observed in the northeast of the study area (Fig. 9).

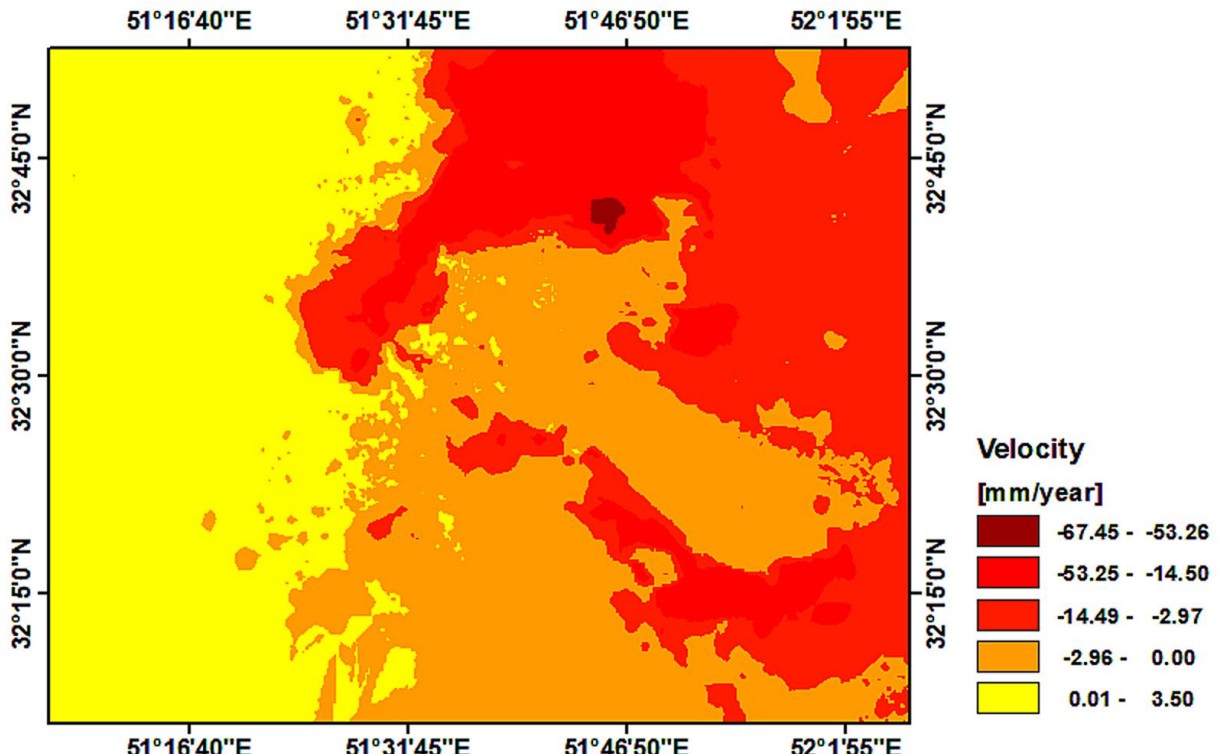


**Figure 9: Velocity map using Kriging**

In this research, in order to obtain a continuous subsidence surface of a specific area, two methods, Kriging and CNN,
Kriging method is based on mathematics and interpolation between cumulative displacement of PSs. The maximum
amount of cumulative displacement obtained by the Kriging method in the studied area is approximately 36 cm (Fig.

401    10).

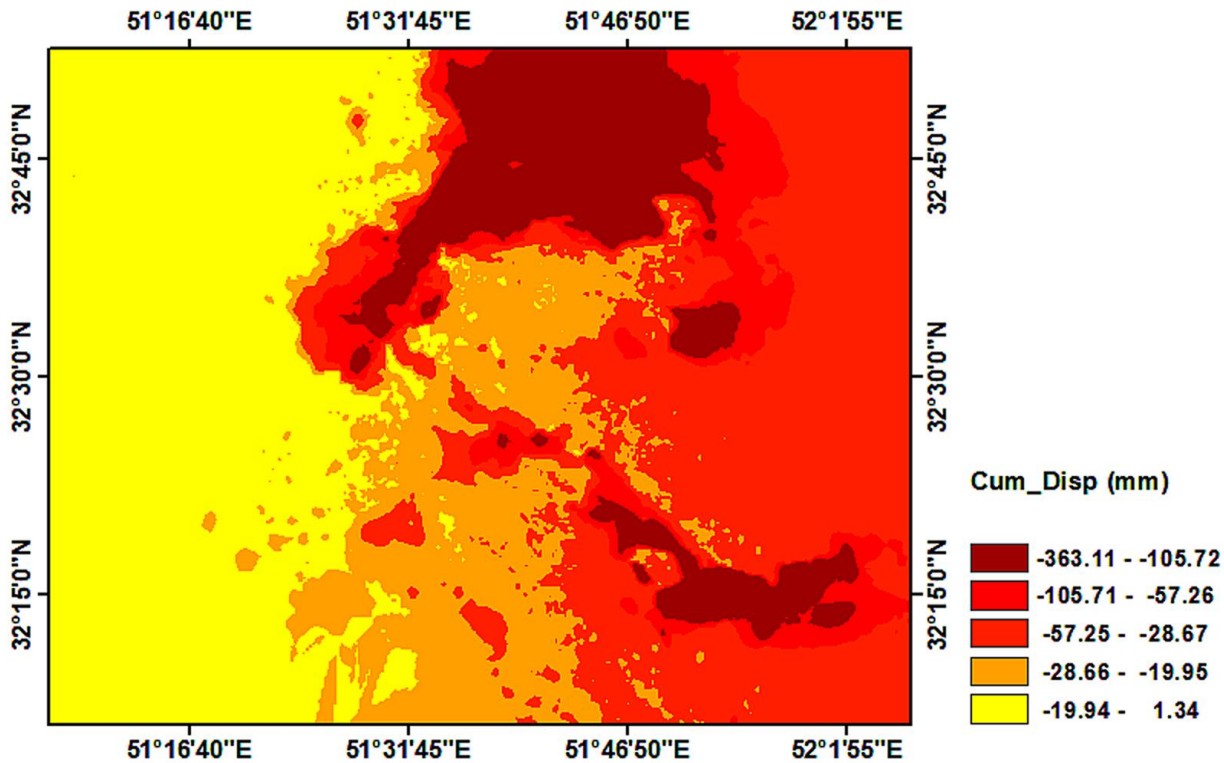


**Figure 10: Cumulative displacement using Kriging in 2014 to 2020**
The CNN method was trained with the cumulative displacement of PSs and the subsidence driving forces in these
points, and finally the subsidence of the entire area was predicted with this model. The maximum amount of
cumulative displacement obtained by the CNN method in the studied area is approximately 33 cm (Fig. 11).

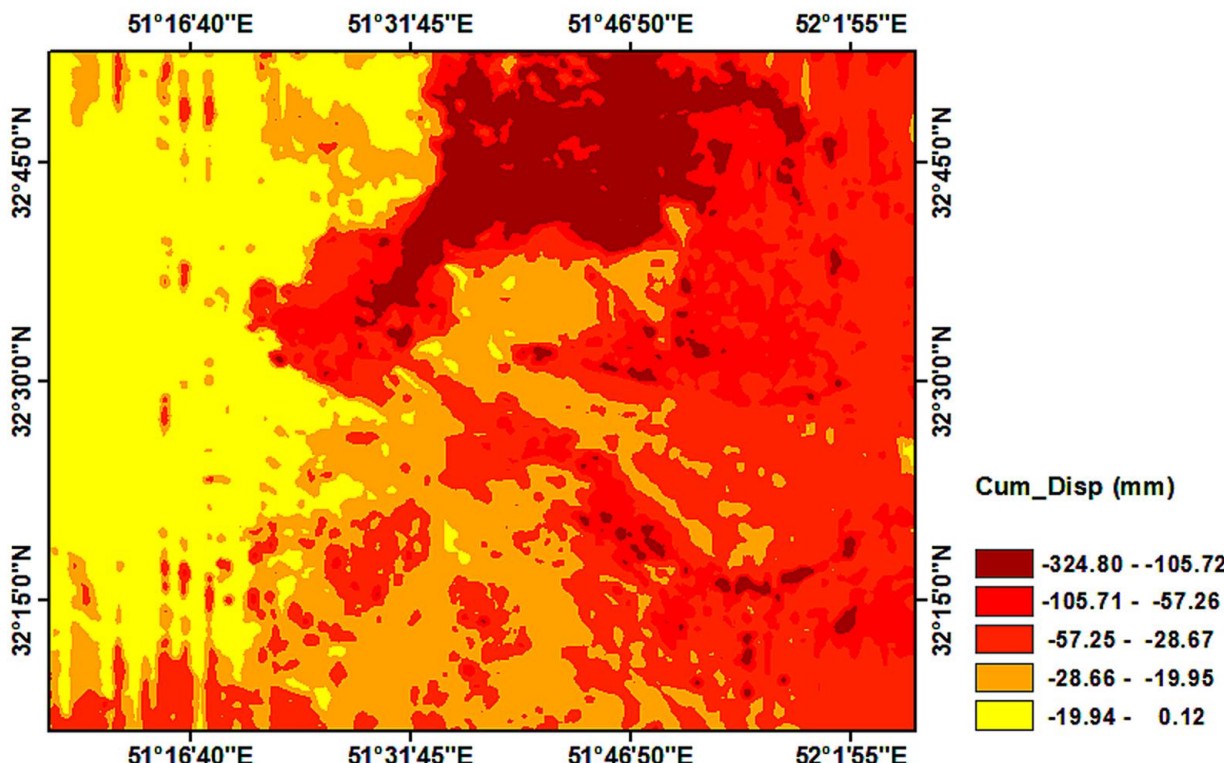


**Figure 11: Cumulative displacement using CNN in 2014 to 2020**

Shahid Beheshti Airport in Isfahan is experiencing a critical rate of land subsidence, with an estimated velocity

exceeding 45 mm per year. This alarming rate of deformation has resulted in a significant cumulative displacement of

approximately 41 cm between 2014 and 2020. Moreover, the CNN-generated subsidence map reveals a slightly higher

maximum cumulative displacement of 42 cm in the region, suggesting that deep learning models provide a more

comprehensive and accurate representation of land deformation. These findings highlight the urgency of addressing

subsidence-related risks, particularly in critical infrastructure areas such as airports, where even slight ground

movements can lead to substantial damage. The CNN model's ability to detect and quantify subsidence in regions

with sparse data further underscores its potential as a valuable tool for monitoring and mitigating land deformation

across various urban and industrial settings (Fig. 12).















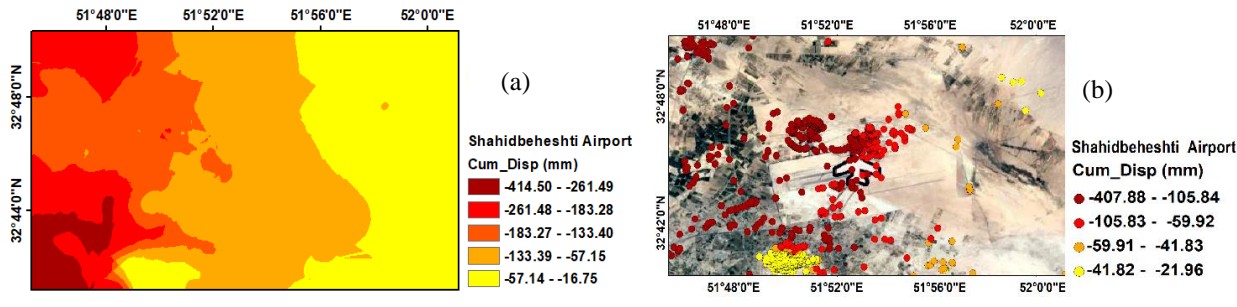

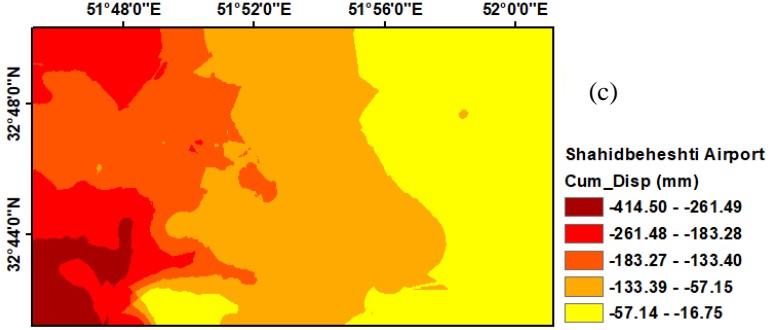

**Figure 12: Cumulative Displacement of Shahid Beheshti Airport in 2014 to 2020: (a) Continuous surface of cumulative displacement using Kriging interpolation between PSs (b) Cumulative displacement of PSs (c) Continuous surface of cumulative displacement resulting from CNN**

Our study revealed significant subsidence patterns in the Mahyar and Nasr Abad Jarqouye regions, highlighting the severity of land deformation over the observation period. The analysis indicates an average subsidence velocity of approximately 5 cm per year, leading to a substantial cumulative displacement of around 33 cm between 2014 and 2020. When applying the Kriging interpolation method, the estimated maximum cumulative displacement reached approximately 35 cm. In contrast, our deep learning-based CNN model predicted a slightly lower maximum cumulative displacement of around 32 cm. These findings underscore the variations in prediction accuracy between traditional geostatistical methods and data-driven deep learning approaches. The discrepancy between Kriging and CNN estimates suggests that while Kriging may slightly overestimate extreme displacement values due to its spatial smoothing effect, the CNN model, trained directly on observed deformation patterns, offers a more data-driven approach to subsidence prediction (Fig. 13).

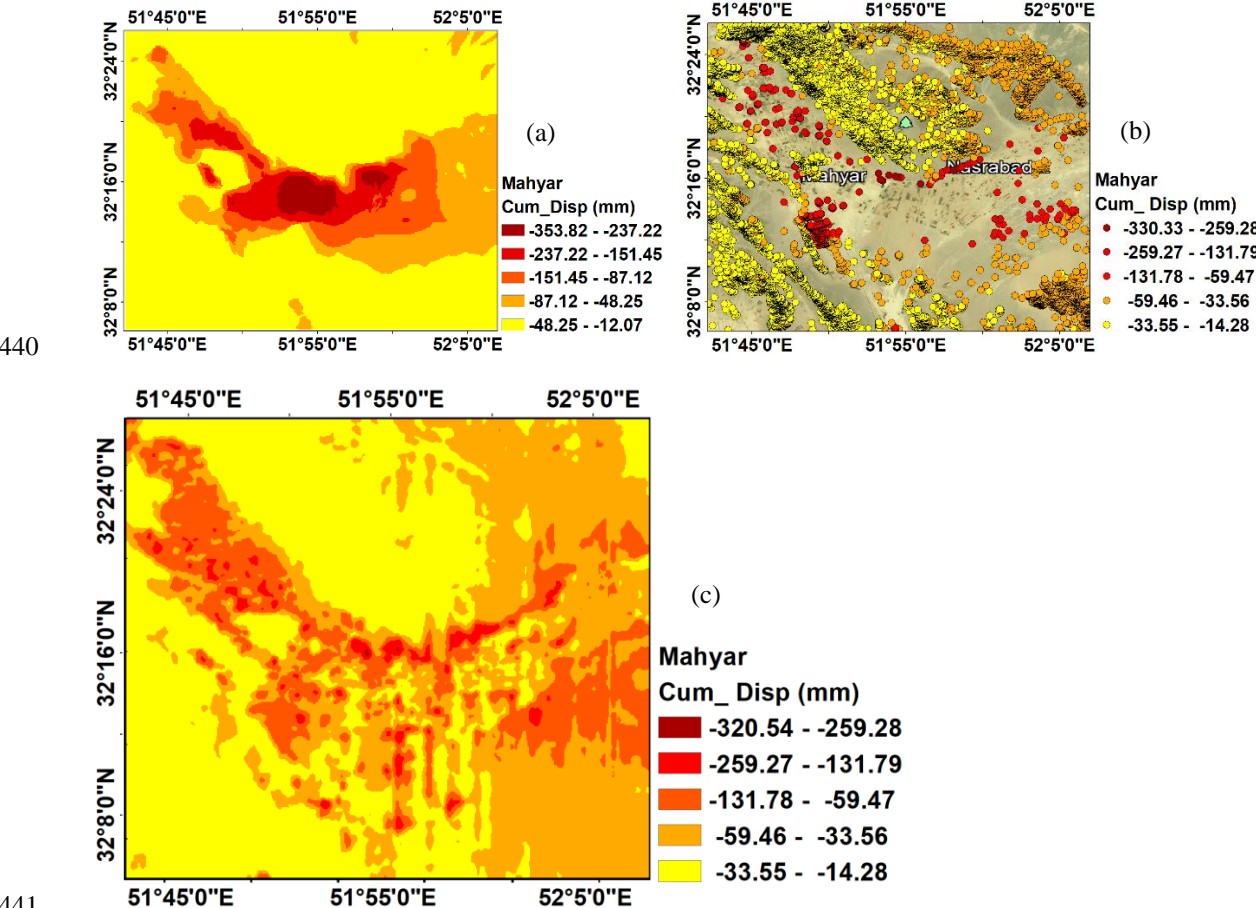

**Figure 13: Cumulative Displacement of Mahyar and Nasr Abad Jarqouye in 2014 to 2020: (a) Continuous surface of cumulative displacement using Kriging interpolation between PSs (b) Cumulative displacement of PSs (c) Continuous surface of cumulative displacement resulting from CNN**

In the Naqsheh Jahan area, the maximum cumulative displacement estimated using the Kriging and CNN methods between 2014 and 2020 was approximately 6 cm and 12 cm, respectively. Similarly, in the Si-o-Se Pol area, the Kriging method estimated a maximum cumulative displacement of around 6 cm, while the CNN predicted a significantly higher value of approximately 19 cm. These discrepancies highlight fundamental differences between geostatistical interpolation and deep learning-based predictive modeling. While Kriging interpolation effectively fits observed PSs, it struggles with accurate extrapolation in regions where measurement points are sparse or absent. Conversely, CNN approach identifies significant deformation trends that Kriging fails to detect, emphasizing the potential of deep learning techniques for more reliable and spatially comprehensive subsidence prediction (Fig. 14).

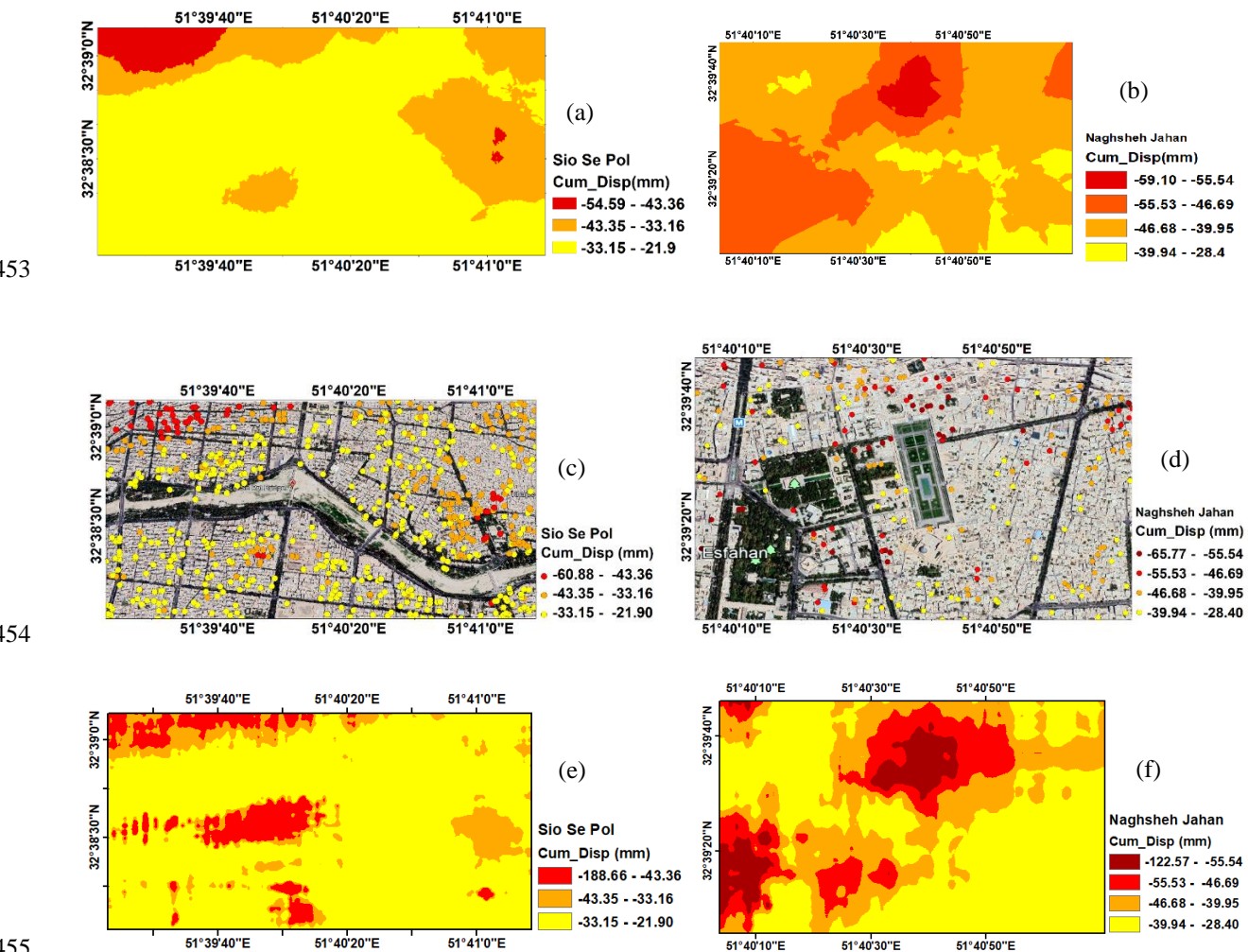

453

454

455

**Figure 14: Cumulative Displacement of Naghsheh Jahan and Si-o-Se Pol area, 2014 to 2020: (a), (b) Continuous surface of cumulative displacement using Kriging interpolation between PSs (c), (d) Cumulative displacement of PSs (e), (f) Continuous surface of cumulative displacement resulting from CNN**

The city of Khomeini Shahr is facing a concerning situation where the velocity has been estimated to be more than 45 mm per year. Unfortunately, this has resulted in displacement in residential areas, with the maximum cumulative displacement of PSs reaching 30 cm from 2014 to 2020. According to the map generated using CNN, the maximum cumulative displacement is currently at 31 cm. A comparative analysis of Kriging interpolation and the CNN model against PSInSAR observations reveals key methodological differences. The Kriging interpolation method, while effective in fitting observed data points, primarily relies on mathematical interpolation and spatial smoothing. This often leads to inaccuracies in regions with a lower density of PS points, as it lacks the ability to infer displacement patterns beyond the available observations. In contrast, the CNN model estimates settlement values based on learned structural relationships, capturing complex spatial dependencies and underlying deformation mechanisms more

effectively. This advantage allows the deep learning model to provide a more continuous and spatially coherent
subsidence map, improving predictive accuracy in areas with sparse measurement data (Fig. 15).

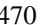

**Figure 15: Cumulative Displacement of Khomeini Shahr, 2014 to 2020: (a) Continuous surface of cumulative displacement**
**using Kriging interpolation between PSs (b) Cumulative displacement of PSs (c) Continuous surface of cumulative**
**displacement resulting from CNN**
In Falavarjan city, the estimated subsidence velocity exceeds 23 mm per year, highlighting a concerning rate of land
deformation. Over the period from 2014 to 2020, this has resulted in a maximum cumulative displacement of
approximately 16 cm based on conventional geostatistical estimates. However, the CNN-generated subsidence map
indicates a significantly higher maximum cumulative displacement of around 23 cm. The discrepancy between
PSInSAR estimates and CNN predictions highlights fundamental differences in their modeling approaches. While
conventional methods rely on spatial interpolation and statistical assumptions, CNNs leverage spatial dependencies
and structural patterns learned from observed data, allowing for more accurate and continuous subsidence mapping.
This suggests that deep learning-based approaches may provide a more reliable representation of ground deformation,
particularly in regions with a sparse distribution or absence of PSs (Fig. 16).

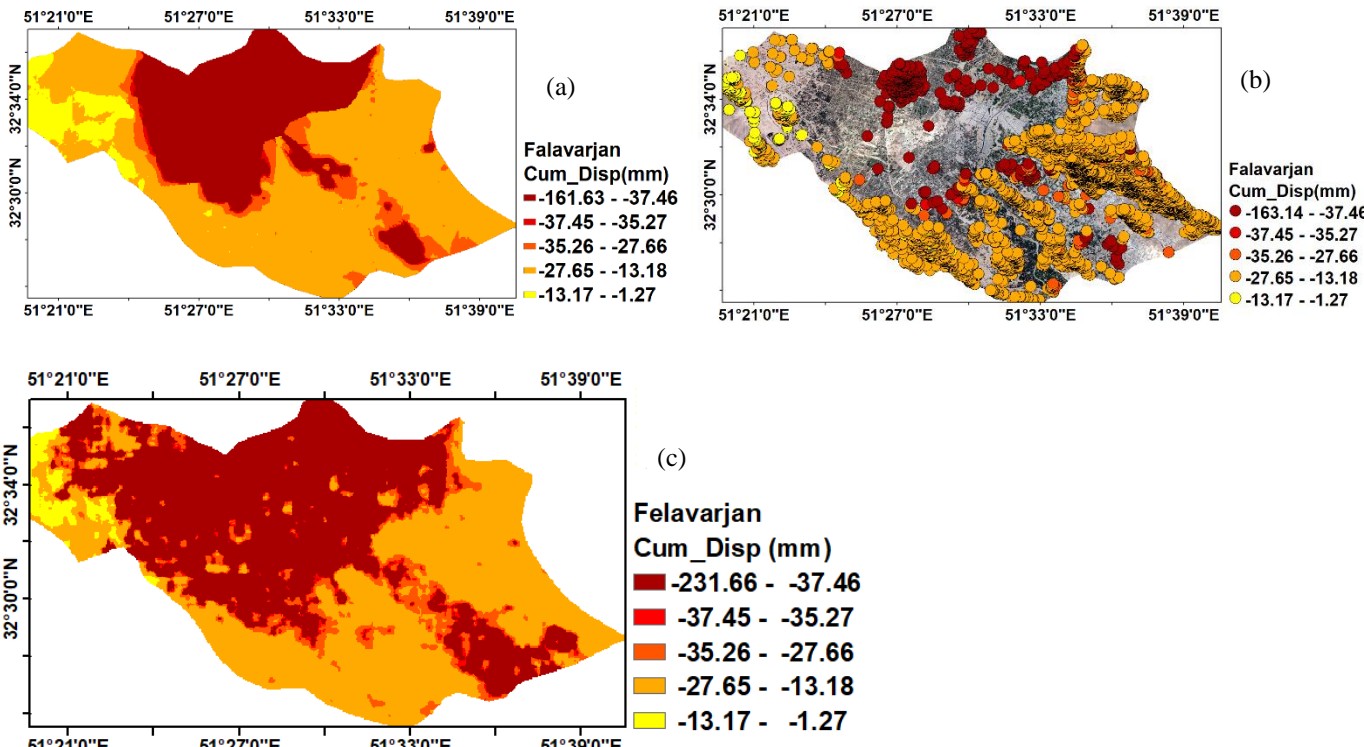

**Figure 16: Cumulative Displacement of Falavarjan, 2014 to 2020: (a) Continuous surface of cumulative displacement using Kriging interpolation between PSs (b) Cumulative displacement of PSs (c) Continuous surface of cumulative displacement resulting from CNN**

**Data Availability**
The data used in this study consists of subsidence measurements obtained from Sentinel-1A and Landsat 8 images
over the period of 2014-2020. The subsidence was calculated using the Sarproz and driving forces of subsidence was
calculated using the ENVI software tools. Sentinel-1A Data: The Sentinel-1A images were used to Calculation of
subsidence through PSInSAR in Sarproz (Version [pcodes_2019-10-02]). Landsat 8 Data: The Landsat 8 images were
used to calculate Land use and NDVI using ENVI (Version [5.3]). Digital Elevation Model: DEM was used to
calculate TWI, SPI, Aspect, Slope, Altitude  using ArcGIS (Version [10.7.1]

**5. Conclusion**

This study presents an innovative deep learning framework utilizing a Convolutional Neural Network (CNN) to generate a continuous subsidence surface across the study area. Unlike traditional methods that rely on discrete geodetic measurements, the proposed approach integrates multiple key driving factors—including NDVI, distance from wells, land use, water table depth, altitude, slope, SPI, TWI, and aspect—providing a more comprehensive and data-driven understanding of subsidence dynamics. The CNN model effectively addresses the limitations of PSInSAR, which, despite its reliability in detecting gradual land deformation, is restricted to persistent scatterers (PSs) and performs poorly in vegetated or low-coherence areas. By leveraging deep learning, the proposed model enables subsidence estimation even in regions where PSInSAR measurements are unavailable, addressing a critical gap in geospatial monitoring.

The superiority of the CNN-based approach was demonstrated through a comparative analysis against conventional interpolation techniques, including Kriging, IDW, and RBF. The CNN model achieved significantly lower RMSE values (3.99 mm, 8.47 mm, and 9 mm for the training, validation, and test datasets, respectively) and an $R^2$ score of 0.98, whereas traditional methods exhibited considerably higher RMSE values (Kriging: 61.60 mm, IDW: 66.21 mm, RBF: 61.76 mm) and negative $R^2$ scores, highlighting their limitations in subsidence prediction. The study also identified severe land subsidence in key areas, with rates exceeding 45 mm per year at Shahid Beheshti Airport and over 54 mm per year in the Mahyar Plain. The CNN model demonstrated an 85% improvement in prediction accuracy over traditional methods, underscoring its robustness and effectiveness, particularly in areas with sparse and irregularly distributed data.

Despite these advancements, some challenges remain. The model's performance is influenced by the availability and quality of input data, and its computational demands necessitate high-performance GPUs for efficient training. Additionally, regional variations in subsidence mechanisms may require model adaptations to ensure accuracy across diverse landscapes. Future research should focus on enhancing the model's generalizability across different geographical regions, developing real-time monitoring capabilities for early warning systems, and integrating additional datasets—such as climate variables and bedrock depth—to further refine predictive accuracy. Furthermore, exploring hybrid deep learning architectures, such as CNN-LSTM models, may enhance computational efficiency and improve temporal prediction capabilities. Addressing these aspects will further establish deep learning-based

subsidence modeling as a scalable and effective tool for geospatial analysis, environmental monitoring, and urban planning.

**Code and data availability**

The Excel file in the Zenodo repository contains 62,000 data points corresponding to permanent scatterers obtained from the PSInSAR method. The nine satellite images used as inputs for the model, which include NDVI, Landuse, etc., were calculated using Landsat 8 and DEM images from the area. These images are also available in the Zenodo repository. Additionally, the Python code for the CNN model used in this paper are accessible through the Zenodo archive at the following link: https://zenodo.org/records/12721120 (Azarm, 2024).

**Author contribution**

Azarm contributed to the writing of the manuscript and conducted the analytic calculations and numerical simulations with the support and supervision of Mehrabi and Nadi. All authors actively participated in discussing the results, providing comments on the manuscript, and revising the final version.

**Competing interests**

The contact author has declared that none of the authors has any competing interests.

**Acknowledgment**

The authors gratefully acknowledge the European Space Agency for providing the Sentinel-1 datasets of the Copernicus mission, which were indispensable for this study. They also extend their sincere appreciation to the Isfahan Regional Water Organization for providing the piezometer data. The data were processed using SARPROZ (Copyright (c) 2009-2015 Daniele Perissin) and visualized in Matlab®, with the support of Google Maps and Google Earth. Additionally, the authors would like to express their gratitude to the US Geological Survey for making the SRTM 1 Arc-Second Global DEM data available, which played a crucial role in the data processing and analysis presented in this paper.

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
