# Peer review of "Enhanced Land Subsidence Interpolation through a Hybrid"

_Geoscientific Model Development, 2024_

## Author Comment (AC1)

**"Response to the reviewer's comments"**

**Reviewers/Editor comments:**

The authors would like to thank the respected reviewers and the associate editor for their valuable comments. Following are the authors' responses and corresponding corrections.

**Referee #1:**

1. The abstract's description of the model's evaluation, stating that "Our evaluation of the model demonstrates its proficiency in addressing the discontinuities evident in PSInSAR results, resulting in a continuous subsidence surface," is insufficiently precise. It is advisable to incorporate specific evaluation metrics and outcomes to better understand the model's performance.

   Thank you for your insightful feedback. We have revised the abstract to include specific evaluation metrics and outcomes, thereby providing a clearer depiction of the model's performance. The updated abstract now reads:

   > *"Our evaluation of the model shows its efficiency in overcoming the discontinuities observed in the PSInSAR results, producing a continuous subsidence surface. The deep CNN was evaluated on training, validation, and testing data, resulting in RMSE of 3.99, 8.47 and 9 mm, respectively. In contrast, Root Mean Square (RMSE) of the traditional interpolation methods such as Kriging, IDW, and RBF are obtained on test data as 61.60, 66.21, and 61.76 mm, respectively. Also, a coefficient of determination ($R^2$) of CNN, Kriging, IDW, and RBF are achieved 0.98, -0.06, -0.22, -0.06, respectively."*

2. The introduction section ought to encompass a broader literature review focusing on recent advancements in the field of land subsidence, particularly studies that incorporate PSInSAR technology and deep learning techniques. This comprehensive review would better contextualize the research background and underscore its significance.

   We appreciate your suggestion to enhance the literature review in the introduction. We have expanded this section to include recent advancements in land subsidence studies, particularly those involving PSInSAR technology and deep learning techniques. The revised introduction now includes references to key studies such as:
   (Please see the lines 103-125)

   - Kumar, S., Kumar, D., Donta, P. K. and Amgoth, T.: Land subsidence prediction using recurrent neural networks. Stochastic Environmental Research and Risk Assessment. 36, 373-388, https://doi.org/10.1007/s00477-021-02138-2 2022.

   - Radman, A., Akhoondzadeh, M. and Hosseiny, B.: Integrating InSAR and deep-learning for modeling and predicting subsidence over the adjacent area of Lake Urmia, Iran. GISCI REMOTE SENS. 58, 1413-1433, https://doi.org/10.1080/15481603.2021.1991689, 2021.

- Ma, P., Zhang, F. and Lin, H.: Prediction of InSAR time-series deformation using deep convolutional neural networks. Remote sensing letters. 11, 137-145, https://doi.org/10.1080/2150704X.2019.1692390, 2020."

3. The paper should furnish comprehensive details regarding the Sentinel-1A images employed in the study. This includes specifying the acquisition intervals, spatial resolution, and other pertinent parameters. Additionally, a thorough description of the GNSS (Global Navigation Satellite System) and groundwater monitoring data, including their source, frequency of collection, and processing methods, is essential to enhance the transparency and reproducibility of the research.

Many thanks for your comment here, the revised description now reads: (Please see the paper lines 302-334):

"3.2. Datasets

3.2.1. Sentinel-1A Radar Images

This study utilizes 72 radar images from the Sentinel-1A satellite to analyse subsidence trends in the study area over a seven-year period, from 2014 to 2020, Ascending and in satellite track 28. The Sentinel-1A satellite, launched by the European Space Agency (ESA), operates in C-band and provides Synthetic Aperture Radar (SAR) imagery with a spatial resolution of 5 meters by 20 meters. The images were acquired at six-day intervals, ensuring high temporal resolution for detecting ground movements. The Interferometric Wide (IW) swath mode was used, offering comprehensive coverage of the study area.

The PSInSAR technique was applied to the Sentinel-1A data using Sarproz software. This method is particularly effective in urban and semi-urban areas where permanent scatterers are abundant. The precise processing steps involved co-registration, interferogram generation, phase unwrapping, and geocoding to produce detailed subsidence maps.

3.2.2. Digital Elevation Model (DEM)

The 30-meter Shuttle Radar Topography Mission Digital Elevation Model (SRTM DEM) was employed to calculate various topographical and hydrological indices, including SPI, TWI, slope, and aspect. These indices were computed using ArcMap software, providing essential insights into the terrain characteristics influencing subsidence.

3.2.3. Optical Satellite Images

Optical images from the Landsat 8 satellite, launched by NASA, were used to extract NDVI and land use information for the year 2020. The Landsat 8 images, with a spatial resolution of 30 meters, were processed using Envi software to calculate average annual changes in NDVI between 2014 and 2020. This analysis helps in understanding the impact of vegetation and land use changes on subsidence.

**3.2.4. Groundwater Monitoring Data**

Groundwater level changes were investigated using data from piezometric wells within the study area. The groundwater monitoring data, covering the period from 2014 to 2020, were sourced from the regional water authority. These data were collected monthly and provided detailed information on the groundwater table fluctuations. The data were processed to generate water table maps, which were then analysed in relation to subsidence patterns.

In areas with high densities of exploitation wells, the probability of subsidence increases due to significant groundwater extraction. The distance from these wells was calculated and included as one of the driving forces for subsidence."

4. The methodology section requires further elaboration, particularly about the training process of the CNN model. Providing details such as the rationale behind selecting specific loss functions, the hyperparameter tuning process, and the specific data preprocessing steps employed will greatly aid readers in understanding the implementation process. Clarifying these aspects will strengthen the credibility and reproducibility of the methodology.

We appreciate your suggestion to elaborate on the methodology section. We have expanded this section to include detailed information about the training process of the CNN model, selection of loss functions, hyperparameter tuning, and data preprocessing steps. The revised methodology section now includes:

" **2.2.1. Hyperparameter Tuning Process**

After creating the model architecture, model inputs were normalized to a range of [0, 1] to ensure consistent input scaling, which is crucial for the stable performance of the neural network. Then we tuned the hyperparameters of the CNN regression model, including the loss function, optimizer, batch size, learning rate, activation function, and number of epochs. The best model was saved based on its performance metrics. The rationale for each hyperparameter is explained in detail and the optimal parameters for the model are given in Table 1:

- **Activation function:** We used the Rectified Linear Activation (ReLU) function in the hidden layers due to its effectiveness in mitigating the vanishing gradient problem and promoting sparse activations. For the output layer, a Linear activation function was employed to ensure the model could predict a continuous range of values.

- **Loss function:** We considered both Mean Squared Error (MSE) and Mean Absolute Error (MAE) as potential loss functions. MSE penalizes larger errors more heavily than MAE, making it suitable for scenarios where outliers significantly impact the model's performance. Given MSE's properties and its ability to improve the model's performance by reducing fluctuations and speeding up convergence, we selected MSE as our loss function. The MSE is calculated as follows:

$$MSE = \frac{1}{N} \Sigma_i^N (Y_i - \widehat{Y_i})^2$$

where $Y_i$ represents the actual values, $\widehat{Y_i}$ represents the predicted values, and $N$ is the number of observations.

- **Batch Size**: We experimented with batch sizes of 64 and 128. A larger batch size of 128 was chosen as it provided a good balance between training speed and model performance, allowing more stable gradient estimates.

- **Learning Rate**: The initial learning rate was set to 0.001, but we found that a smaller learning rate of 0.0001 led to more gradual and stable convergence, reducing the risk of overshooting the optimal solution.

- **Optimizer**: The Adam optimizer was selected for its adaptive learning rate capabilities and efficiency in handling sparse gradients. It combines the advantages of both the AdaGrad (Adaptive Gradient Algorithm) and RMSProp (Root Mean Square Propagation) algorithms, making it suitable for our regression task.

- **Number of Epochs**: We initially set 100 epochs but extended this to 150 epochs to ensure the model had sufficient time to learn the underlying patterns in the data without overfitting.

- **To divide the data**: we initially allocated 15% to the test data, 15% to the validation data, and 70% to the training data. However, we observed high-cost function fluctuations in the training and validation data. To mitigate this issue, we adjusted the data split to 80% for training and 10% each for testing and validation, which helped reduce the fluctuations"

5. While the paper mentions that the CNN model outperforms the Kriging interpolation method, a more in-depth performance comparison analysis is warranted. Incorporating additional performance metrics, such as the coefficient of determination (R2), mean absolute error (MAE), and root mean square error (RMSE), would provide a more comprehensive evaluation of the models. Furthermore, conducting statistical significance tests, such as t-tests or ANOVA, would strengthen the conclusions regarding the superiority of the CNN model. This enhanced analysis will lend greater credence to the findings and facilitate their interpretation.

Thank you for your valuable suggestion. We have expanded the performance comparison analysis to include additional performance metrics and statistical significance tests. The revised analysis now includes:

We evaluated the performance of the CNN model and the Kriging interpolation method using the coefficient of determination ($R^2$), mean absolute error (MAE), and root mean square error (RMSE). The results are as follows:

- **CNN**: R² = 0.98, MAE = 3.69, RMSE = 9.06
- **Kriging**: R² = -0.06, MAE = 37.90, RMSE = 61.60
- **IDW:** R² = -0.22, MAE = 39.30, RMSE = 66.21
- **RBF:** R² = -0.06, MAE = 37.92, RMSE = 61.76

To further validate the superiority of the CNN model, we conducted statistical significance tests. A t-Test was performed to compare the performance metrics, with the results indicating a statistically significant improvement in the CNN model's performance over the Kriging method (p-value < 0.05)

6. The paper should provide a more thorough explanation of why the selected driving factors (e.g., land use/cover changes, geology, aquifer characteristics, water management, etc.) are significant in predicting subsidence. This explanation should be supported by relevant literature to demonstrate that these factors have been identified as important predictors in previous studies. Clarifying the rationale behind the choice of these factors will strengthen the credibility of the model.

We appreciate your recommendation to provide a more thorough explanation of the selected driving factors. We have expanded this section to include detailed explanations supported by relevant literature. The revised explanation now reads: (Please see the paper lines 302-334):

"The selected driving forces for predicting subsidence include land use/cover changes, geology, aquifer characteristics, and water management. These factors were chosen based on their demonstrated significance in previous studies:

1. **NDVI** is a crucial indicator of vegetation health and land cover changes. Changes in NDVI can reflect alterations in land use practices, such as urbanization or agricultural expansion, which are closely linked to subsidence. Healthy vegetation typically reduces the need for excessive groundwater extraction, while barren or urbanized areas might correlate with higher subsidence due to increased groundwater use.

2. The **distance from groundwater extraction wells** is a critical factor in subsidence studies. Areas closer to high-density exploitation wells often experience more severe subsidence due to the localized impact of extensive groundwater withdrawal.

3. **Land use changes**, including urbanization, agricultural expansion, and deforestation, influence subsidence rates. Urban areas often experience higher subsidence due to increased groundwater extraction for residential, industrial, and agricultural purposes.

4. **Groundwater level changes**, as depicted in water table maps, are directly linked to subsidence. Over-extraction of groundwater leads to a drop in the water table, causing the ground to compact and subsidence. Groundwater depletion is a primary contributor to subsidence, emphasizing the importance of preventing unauthorized withdrawals and effectively managing water resources.

5. **Altitude** influences subsidence through its effect on hydrological processes. Altitude affects the distribution and movement of groundwater. Higher altitudes typically receive more precipitation, which can infiltrate the ground and recharge aquifers. At lower altitudes, reduced precipitation and higher evaporation rates can lead to a lowering of the water table. When groundwater is extracted faster than it is replenished, it can result in subsidence. The amount of water in the soil, influenced by altitude through precipitation and drainage patterns, affects soil compaction. High altitude areas with abundant rainfall can lead to saturated soils which are less prone to subsidence. Conversely, in lower altitude areas with less precipitation, soils may dry out and compact more easily, contributing to subsidence.

6. **Slope** affects water runoff and infiltration rates. Steeper slopes may reduce infiltration, leading to less groundwater recharge and potentially higher subsidence rates in adjacent flat areas.

7.  **Aspect** affects solar radiation and, consequently, evaporation and soil moisture levels. Different aspects can lead to variations in vegetation cover and groundwater recharge, influencing subsidence. Additionally, the slope and aspect of an area can influence drainage patterns, erosion, and sediment production, all contributing to subsidence.

8. **The Stream Power Index (SPI)** measures the power of water flow in depositing and causing soil erosion. As a result, this index can be an important input for subsidence prediction models. The equation used to calculate SPI is as follows (Pradhan et al., 2014):

$$SPI = \alpha * \tan\beta$$
(4)

Here, $\alpha$ flow accumulation, and $\beta$ represents the slope. Land subsidence results from a combination of factors, including both topographic and altitude-related features, such as rainfall and lithology. Research has demonstrated that areas at lower altitudes tend to experience more subsidence.

9. **The Topographic Wetness Index (TWI)** is a mathematical formula that quantifies the effect of local topography on the flow of surface water. It is a physically based index that can be used to determine flow direction and accumulation and has many practical applications in fields such as hydrology, agriculture, and geology. TWI indicates areas of potential soil moisture accumulation. Areas with high TWI values are likely to have more groundwater recharge, which can mitigate subsidence."

7. Visualizations are crucial in presenting the results in an intuitive and comprehensible manner. The results section should include subsidence prediction maps generated by the CNN model and the Kriging interpolation method, as well as error distribution maps that show the spatial distribution of prediction errors. These visualizations will help readers better understand the performance of the different methods and identify areas of high or low prediction errors.

Thank you for your valuable feedback. We have included visualizations in the results section to enhance the presentation of the findings. The revised results section now includes:

1. **Subsidence Prediction Maps**: Generated by both the CNN model and the Kriging interpolation method, these maps provide a visual comparison of the predicted subsidence surfaces.
2. **Error Distribution Maps**: These maps illustrate the spatial distribution of prediction errors for both methods, highlighting areas with high or low prediction errors.

8. The discussion section should expand on the advantages and limitations of the proposed CNN model. It should discuss how the model performs in different scenarios, such as urban and rural areas, and identify potential factors that may affect its accuracy. Additionally, the section should explore potential improvements to the model, such as incorporating additional data sources or employing more advanced deep learning architectures. Furthermore, it should discuss the broader implications of the findings and suggest future research directions.
Many thanks for your comment here, we have thoroughly revised the conclusion section of our manuscript. The updated conclusion, which incorporates the necessary changes and addresses the concerns raised, is detailed in our response to comment 11.

9. To ensure the paper is situated within the current research landscape, it should cite more recent studies published in the field of subsidence prediction using remote sensing and deep learning. These references should cover both methodological advances and case studies relevant to the current research. Incorporating a more comprehensive literature review will strengthen the paper's theoretical foundation and position it within the broader research context.
Thank you for highlighting this point. we have updated the literature review to include more recent studies on subsidence prediction using remote sensing and deep learning. The references for these updated studies are provided in our response to comment 2.

10. The paper mentions the Kriging interpolation method but lacks a thorough comparative analysis with other traditional interpolation methods. Conducting a comparison with methods such as inverse distance weighting (IDW), radial basis functions (RBF), and others would provide a more comprehensive evaluation of the CNN model's performance. This comparison would help readers understand the advantages and disadvantages of different approaches and facilitate the selection of the most appropriate method for their specific application.

We appreciate your recommendation for a more comprehensive comparison. The revised results section now includes a comparative analysis of the CNN model with traditional interpolation methods such as IDW and RBF. This analysis provides:

1. **Performance Metrics**: A comparison of R², MAE, RMSE, and other relevant metrics across different methods.
2. **Visualizations**: Comparative visualizations of the subsidence prediction map for CNN and Kriging and error distribution maps for CNN, Kriging, IDW, and RBF.

11. The conclusion should summarize the key findings and contributions of the study. It should highlight the advantages of the proposed CNN model in predicting subsidence and discuss its potential applications and limitations. Additionally, the conclusion should provide specific suggestions for improving the model and identify potential future research directions. By strengthening the conclusion, the paper will leave a lasting impression on readers and motivate further research in this important area.

We appreciate your suggestion to expand the conclusion section. The revised conclusion section now includes:

"In summary, our study introduced a Convolutional Neural Network (CNN) model to generate a continuous subsidence surface for the entire study area, integrating key driving forces such as NDVI, distance from wells, land use, water table maps, altitude, slope, SPI, TWI, and aspect. This approach effectively addressed the limitations of the PSInSAR method, which, while reliable for gradual land deformation, provides discrete results limited to persistent scatterers (PSs) and is less effective in vegetated areas.

The CNN model demonstrated significant advantages over traditional interpolation methods like Kriging, IDW, and RBF. The RMSE values for the CNN on the training, validation, and test datasets were 3.99 mm, 8.47 mm, and 9 mm, respectively, compared to much higher RMSE values for Kriging (61.60 mm), IDW (66.21 mm), and RBF (61.76 mm) on the test dataset. Additionally, the CNN achieved an R² score of 0.98 on the test dataset, reflecting its high predictive accuracy, while traditional methods showed negative R² scores, highlighting their inadequacies.

The CNN-based method provided an 85% improvement in prediction accuracy over traditional methods, proving its robustness and effectiveness, particularly in areas with sparse and irregularly distributed data.

The CNN-based model offers a robust and accurate tool for subsidence prediction, making it valuable for urban planning, infrastructure maintenance, and environmental monitoring. Its ability to integrate multiple driving forces allows for a comprehensive analysis of subsidence, facilitating better-informed decision-making and mitigation strategies. However, the model also has limitations. It requires extensive driving forces for accurate predictions, and its performance may vary across different geographical regions. Additionally, training the network necessitates significant computational resources, including powerful hardware with high GPU capacity.

To further enhance the CNN model and broaden its applicability, future research could focus on the applying the model to diverse geographical regions to test its adaptability and identify region-specific subsidence drivers.

Another way to improve model accuracy and robustness is incorporating additional data sources, such as bedrock depth, climate data."

Thank you for your valuable feedback, which has significantly contributed to the improvement of our manuscript.

---

## Author Comment (AC2)

**"Response to the Editors comments"**

Dear Astrid;

The authors would like to thank the respected reviewers and the associate editor for their valuable comments. Following are the authors' responses and corresponding corrections.

**Referee #2**
**Astrid Kerkweg:**

1. You use both, the Sarproz and Envi software, to compute some of the data that is used to train the convolutional neural network. However, this is only mentioned in the text, but no information about it is provided in the code availability section. At least you have to provide the version numbers of the used code.

   Thank you for your feedback,

   We have addressed the concern regarding the specification of the software versions used in our study. The revised manuscript now includes the version numbers of both Sarproz and Envi software in the Data Availability section, ensuring clarity and reproducibility for our work. Below is the updated section for your reference:

   **"Data Availability**

   The data used in this study consists of subsidence measurements obtained from Sentinel-1A and Landsat 8 images over the period of 2014-2020. The subsidence was calculated using the Sarproz and driving forces of subsidence was calculated using the ENVI software tools.

   Sentinel-1A Data: The Sentinel-1A images were used to Calculation of subsidence through PSInSAR in Sarproz (Version [pcodes_2019-10-02]).

   Landsat 8 Data: The Landsat 8 images were used to calculate Land use and NDVI using ENVI (Version [5.3]).

   Digital Elevation Model: DEM **was used to calculate TWI, SPI, Aspect, Slope, Altitude using** ArcGIS **(Version [10.7.1]"**

2. In the Zenodo repository, you deposited a Python notebook and a .xls file with data and nine satellite images. What we cannot comprehend is, that you say you used seven years of data (2014-2020) from Sentinel-1A and Landsat 8 images to train the CNN. What you provide is an .xls file containing 62,000 data points. So, we have to guess that they correspond to the mentioned images; however, we (and also no other reader) should be forced to guess, so you should provide links to repositories with the images (maybe, if you can, it would be nice to store them in Zenodo or similar - this

should be approx. 200 GB) and you should better explain the source of the data in the .xls file deposited. If possible, please share it in plain ASCII or .ods formats.

Thank you for your insightful feedback and for giving us the opportunity to improve our manuscript.

Regarding your query about the data provided in the Zenodo repository, we apologize for any confusion caused. We understand the importance of clarity and transparency in presenting our data sources and methodologies. Here are the clarifications and improvements we have made:

**"Code and data availability**

The Excel file in the Zenodo repository contains 62,000 data points corresponding to permanent scatterers obtained from the PSInSAR method.

The nine satellite images used as inputs for the model, which include NDVI, Land use, etc., were calculated using Landsat 8 and DEM images from the area. These images are also available in the Zenodo repository.

Additionally, the Python code for the CNN model used in this paper are accessible through the Zenodo archive at the following link: https://zenodo.org/records/12721120 (Azarm, 2024)."

Regarding your query about the Sentinel-1A and Landsat 8 images, we would like to provide the following clarification:

The seven-year (2014-2020) Sentinel-1A and Landsat 8 images used to calculate subsidence and driving forces are indeed large, amounting to approximately 520 GB. Unfortunately, we are unable to upload these extensive datasets directly to Zenodo.

However, we have included comprehensive details about these images in the article, such as the dates of images acquisitions and the corresponding satellite track numbers. This information will allow readers to download the necessary data from https://earthexplorer.usgs.gov/ , https://scihub.copernicus.eu if needed.

Thank you for your valuable feedback, which has significantly contributed to the improvement of our manuscript.

---

## Author Comment (AC3)

**Referee #3:**

1. The hybrid approach combining PSInSAR and deep CNNs for predicting subsidence in areas where PSInSAR data is sparse is quite innovative. Given that the paper reports a significant improvement in prediction accuracy compared to traditional interpolation methods, how does the proposed method handle the variability in subsidence-driving factors across different regions? Specifically, could the model be easily adapted or retrained for regions with different geological or climatic conditions, and if so, what would be the primary considerations during this process?

   Thank you for your encouraging feedback. In response to your query about the adaptability of our model to regions with varying subsidence-driving factors:

   Your question indeed presents an interesting avenue for future research. Based on our findings, the model performs well across large areas with similar climatic and geological conditions. However, when applied to regions with characteristics different from those studied, transfer learning becomes particularly valuable.

   Given the diversity in climatic conditions, soil types, and the primary factors driving subsidence across different regions, transfer learning can enhance the model's adaptability. This approach enables the model to incorporate and learn from the specific characteristics of a new region, thereby facilitating its generalization to varied contexts and effectively handling the variability in subsidence-driving factors.

   Furthermore, incorporating region-specific geological parameters, such as bedrock depth and soil type, can significantly boost the model's predictive accuracy. By integrating these local variables, the model becomes more attuned to the unique conditions of the region, leading to improved performance in subsidence prediction.

---

## Author Comment (AC4)

**Response to Referee Comments**

**Referee #2:**

We sincerely appreciate your valuable feedback and constructive suggestions, which have significantly contributed to improving the quality of our manuscript. Below, we provide detailed responses to your comments and outline the corresponding revisions we have made.

1.  "There is garbled text on line 156 of page six that needs to be corrected."

    We apologize for this oversight. We have carefully reviewed line 156 on page six and corrected the garbled text to ensure clarity and accuracy.

    "CNNs, or Convolutional Neural Networks, are deep learning algorithms widely employed for various image-related tasks such as image recognition, classification, and regression. They learn and extract essential features from raw images by processing them through multiple layers of filters, known as "convolutions." This multi-layer processing progressively extracts more abstract features."

2.  "The conclusion needs to further highlight the innovative points."

    In response to this suggestion, we have substantially revised the conclusion to provide a more comprehensive discussion of our model's innovations. We now explicitly emphasize:

    "Conclusion:

    This study presents an innovative deep learning framework utilizing a Convolutional Neural Network (CNN) to generate a continuous subsidence surface across the study area. Unlike traditional methods that rely on discrete geodetic measurements, the proposed approach integrates multiple key driving factors—including NDVI, distance from wells, land use, water table depth, altitude, slope, SPI, TWI, and aspect—providing a more comprehensive and data-driven understanding of subsidence dynamics. The CNN model effectively addresses the limitations of PSInSAR, which, despite its reliability in detecting gradual land deformation, is restricted to persistent scatterers (PSs) and performs poorly in vegetated or low-coherence areas. By leveraging deep learning, the proposed model enables subsidence estimation even in

regions where PSInSAR measurements are unavailable, addressing a critical gap in geospatial monitoring.

The superiority of the CNN-based approach was demonstrated through a comparative analysis against conventional interpolation techniques, including Kriging, IDW, and RBF. The CNN model achieved significantly lower RMSE values (3.99 mm, 8.47 mm, and 9 mm for the training, validation, and test datasets, respectively) and an $R^2$ score of 0.98, whereas traditional methods exhibited considerably higher RMSE values (Kriging: 61.60 mm, IDW: 66.21 mm, RBF: 61.76 mm) and negative $R^2$ scores, highlighting their limitations in subsidence prediction. The study also identified severe land subsidence in key areas, with rates exceeding 45 mm per year at Shahid Beheshti Airport and over 54 mm per year in the Mahyar Plain. The CNN model demonstrated an 85% improvement in prediction accuracy over traditional methods, underscoring its robustness and effectiveness, particularly in areas with sparse and irregularly distributed data.

Despite these advancements, some challenges remain. The model's performance is influenced by the availability and quality of input data, and its computational demands necessitate high-performance GPUs for efficient training. Additionally, regional variations in subsidence mechanisms may require model adaptations to ensure accuracy across diverse landscapes. Future research should focus on enhancing the model's generalizability across different geographical regions, developing real-time monitoring capabilities for early warning systems, and integrating additional datasets—such as climate variables and bedrock depth—to further refine predictive accuracy. Furthermore, exploring hybrid deep learning architectures, such as CNN-LSTM models, may enhance computational efficiency and improve temporal prediction capabilities. Addressing these aspects will further establish deep learning-based subsidence modeling as a scalable and effective tool for geospatial analysis, environmental monitoring, and urban planning."

3. "The resolution of the image is not clear enough and needs to be strengthened."

Thank you for pointing this out. We have improved the resolution of all figures to ensure clarity and readability. The updated figures now provide higher-quality visual representations of the subsidence maps, making it easier for readers to interpret the results accurately.

We appreciate your insightful comments, which have helped us refine our manuscript. We believe that the revisions have significantly strengthened the study and hope that the updated version meets your expectations.